## Science, society and policy

human-computer interaction

deepfake, experiments, manual detection

**Author for correspondence:**
Andrew Lewis
e-mail: andrewfranklewis@icloud.com

# Deepfake detection with and without content warnings

Andrew Lewis[1], Patrick Vu[2], Raymond M. Duch[1] and Areeq Chowdhury[3]

[1]University of Oxford, Oxford, UK
[2]Brown University, Providence, RI, USA
[3]The Royal Society, London, UK

 AL, 0000-0002-6224-2828

The rapid advancement of 'deepfake' video technology—which uses deep learning artificial intelligence algorithms to create fake videos that look real—has given urgency to the question of how policymakers and technology companies should moderate inauthentic content. We conduct an experiment to measure people's alertness to and ability to detect a high-quality deepfake among a set of videos. First, we find that in a natural setting with no content warnings, individuals who are exposed to a deepfake video of neutral content are no more likely to detect anything out of the ordinary (32.9%) compared to a control group who viewed only authentic videos (34.1%). Second, we find that when individuals are given a warning that at least one video in a set of five is a deepfake, only 21.6% of respondents correctly identify the deepfake as the only inauthentic video, while the remainder erroneously select at least one genuine video as a deepfake.

## 1. Introduction

Fitting in with wider trends of digital mis- and dis-information, deepfakes have the potential to further fracture the shared basis of truth in an already polarized society. Because of the still-nascent state of the technology, it is difficult to predict how deepfakes of individuals or events may be deployed and received. The possibility of bad actors using this technology for any number of deceptive ends—from *revenge porn* and identity fraud to terrorism and election manipulation—has prompted both the FBI [1] and Europol [2] to issue warnings, with the latter calling deepfakes 'perhaps the most immediately tangible, damaging application' of artificial intelligence (p. 52).

These warnings underscore the dual threat of deepfakes: first, the technology provides a new and potentially persuasive means of spreading false information; second, the difficulty of manually detecting real from fake videos (i.e. with the naked eye) threatens

to lower the information value of video media entirely. As people internalize deepfakes' capacity to deceive, they may place less trust in all online videos, including authentic content [3]. This may also be exacerbated by what is known as the *Liar's Dividend*—that is, genuine videos being written off as deepfakes by those with an interest in discrediting them [4]. The result in both cases is increased uncertainty, which can bolster motivated information processing and belief formation [5,6]. This is why Europol warns, somewhat ominously, that deepfakes could 'undermine the possibility of a reliable shared 'reality' (p. 53).

As concerns about the veracity of videos spread, it may fall on regulators and technology companies to serve as moderators of authenticity. If individuals lack the ability to discern deepfakes from genuine videos manually, detection algorithms could be the only way to consistently and accurately flag fake content. At the time of writing, state-of-the-art detection algorithms achieve roughly 65% accuracy when faced with videos 'in the wild' [7], while Groh *et al.* [8] find human detection rates to be comparable to or slightly better than these tools. Technology companies such as Meta and Google are devoting resources towards honing these technologies, with the former making headlines last year for its new 'model parsing' detection technique, which it predicts may be able to not only detect but also find the provenance of deepfakes in the future [9,10].

Yet even as these technologies advance, it remains unclear what the best approach will be for dealing with deepfakes on prominent social media sites. (For example, one might imagine that false videos intended to incite violence would be treated rather differently than those used for entertainment or satire.) As with text- and photo-based misinformation, potential remedies include labelling deepfakes with content warnings or removing them from platforms entirely. The former approach has gained traction in recent years, with both Facebook and Twitter making efforts to flag fake or misleading content during the COVID-19 pandemic [11].

If content warnings are to be among platforms' preferred strategies for informing users that a video is a deepfake, it is important to interrogate how such warnings affect subsequent perceptions of videos. Our research contributes to the literature on content warnings for misinformation and manual deepfake detection by addressing two questions: first, do people note something out of the ordinary when encountering a deepfake among a group of videos without a prior warning; second, does *a prior* warning that at least one video they will view is a deepfake enable individuals to then scrutinize them more closely and distinguish between authentic and inauthentic videos? The latter is a critical question. If individuals cannot detect deepfakes even with content warnings, it renders faith in a third party's judgements essential: as detailed below, low manual detection ability would necessarily force people to rely on external sources of evaluation if they wish to hold accurate beliefs about the veracity of a specific video. Such is the purpose of this paper: in an experiment described below, we test how likely UK residents are to spot a deepfake from a genuine video in both natural contexts (i.e. without a warning) and when they have been given a content warning.

Our findings in the first treatment show that participants who view videos containing a deepfake with no prior warning are no more likely to spot something out of the ordinary than those who view only authentic videos. This suggests that participants have poor detection abilities when viewing multiple videos in natural settings without prior warnings. In the second treatment, participants are issued a direct warning that at least one of the five videos they will see is a deepfake, and asked to evaluate the veracity of all five videos in the set (which contains one deepfake and four authentic videos). We find participants correctly select the deepfake and only the deepfake in only 21.6% of cases, while nearly half erroneously believe one or more genuine videos to be deepfakes. This rate broadly aligns with past research: while people are better than random at determining whether an individual video is genuine or fake [8], manual detection is nevertheless imperfect, and the likelihood of a classification error increases exponentially with the number of videos one evaluates. Notably, we find that correct detection of deepfakes is uncorrelated with almost all characteristics we observe, including self-reported confidence in detection abilities, familiarity with the actor depicted in the deepfake, gender and level of social media use.

These results show that people have an imperfect ability to classify genuine from fake videos, and thus human discernment alone cannot reliably yield accurate assessments of video veracity. A practical interpretation of the second treatment is that warning labels do not enable individuals to simply look closer and see the irregularities on their own. As such, successful content warnings on deepfakes may rely on trust in external sources of authentication, raising concerns that any such warnings may be discounted as politically motivated or biased.

The rest of this paper proceeds as follows. Section 2 reviews the literatures on content warnings and manual deepfake detection. Section 3 develops a theoretical model examining deepfake detection and content warnings. Section 4 outlines our experimental design, and §5 presents our results. Section 6 concludes with a brief discussion of implications of our findings.

## 2. Literature review

A growing literature assesses the public's capacity to detect deepfakes from genuine videos, and how various conditions—such as the perceived valence of a video or the presence of information about deepfakes—may affect detection. The results of this literature are mixed. This reflects, at least in part, heterogeneity in experimental design as well as the quality, content, and context of the videos presented to study participants. Groh *et al*. [8] find in two separate settings that human subjects perform substantially better than random guessing and in some cases similar to, or better than, leading detection algorithms (whose accuracy rates are around 65%). Similarly, Groh *et al*. [8] show that deepfake identification for political speeches by well-known politicians is better than random chance when people are asked directly. On the other hand, Ternovski *et al*. [12] show that participants are unable to discriminate between authentic and inauthentic videos of a nearly identical pair of politicians (one an actor, one a deepfake rendering of the actor) issuing a statement. Similarly, Vaccari & Chadwick [13] find that only half of participants who watched a deepfake of former President Obama making a 'highly improbable' statement were able to discern the video as untrue. Studying how deepfakes may be deployed by political operatives to undermine their opponents, Dobber and co-authors [14] find that deepfakes that are in line with individuals' worldviews are both likely to be perceived as genuine, and also negatively affect respondents' perceptions of the political opponents in question. In the studies above, some participants were issued a specific warning that they would view a deepfake. The first experimental treatment in this paper considers whether participants are apt to spot something out of the ordinary when encountering deepfakes in a natural setting, without any prior warning. This experimental design is intended to match users' experiences when viewing multiple videos on online platforms that do not issue warnings about the veracity of most content.

Another strand of the literature examines the impact of deepfakes on mistrust of media more generally. Fallis [3] argues that if people become aware of deepfakes but are unable to spot them from real videos with reliable accuracy, they may simply become more sceptical of videos writ large, irrespective of veracity. Some early experimental evidence lends credence to this theory. For example, a generalized warning about the existence of deepfakes (e.g. advances in artificial intelligence have enabled the creation of deepfakes: that is, fake videos that look real) was found to have 'increase[d] disbelief in accompanying video clips—regardless of whether the video is fake or real' ([12], p. 9). This echoes past research on misinformation in print media, which finds general misinformation warnings can reduce trust in both true and false headlines and articles [15,16]. However, at least some of this effect can be mitigated by using specific, rather than general, content warnings [15,17]. We contribute to the synthesis of these literatures by examining the effect of specific warnings for deepfakes on individuals' beliefs about the veracity of both genuine and fake videos.

## 3. Theoretical model

In this section, we develop a simple Bayesian model for understanding the potential impact of content warnings. Consider a viewer watching a video $V$ that is either authentic ($V = 1$) or inauthentic ($V = 0$). Before watching the video, the viewer has *a prior* subjective probability $\pi_F$ that the video is inauthentic. After watching a video, the viewer forms beliefs about whether the video is authentic ($B = 1$) or inauthentic ($B = 0$). This judgement may be based on the characteristics of the specific viewer and video in question, and vary, for example, with the quality of the video, whether it is in fact authentic or inauthentic, or whether it contains a content warning.

We are interested in the probability that a video is inauthentic when it is believed to be inauthentic by the viewer—that is, the probability that the viewer has correctly detected the deepfake. Define $d_{T|T} \equiv P(B = 1 | V = 1)$ as the probability that the viewer believes the video is authentic after viewing it, conditional on it being authentic; and $d_{F|F} \equiv P(B = 0 | V = 0)$ as the probability that the viewer believes the video is inauthentic when it is in fact inauthentic. The quantities $d_{T|T}$ and $d_{F|F}$ measure the ability

of the viewer to accurately detect authentic and inauthentic content. Suppose that these detection rates are functions of whether or not the video contains a content warning, which is denoted by a binary variable $C$. Then the probability of being correct when believing a video to be inauthentic for a Bayesian decision-maker is given by

$$P(V = 0 \mid B = 0) = \frac{d_{F|F}\,(C)\,\pi_F}{d_{F|F}\,(C)\,\pi_F\; + (1 -\; d_{T|T}\,(C))(1 - \pi_F)}.$$

From this simple model, we can derive a number of predictions about the impact of deepfakes and the effectiveness of content warnings.

First, the model highlights the importance of detection abilities. The probability of holding correct beliefs about an inauthentic video is increasing in both the improved detection of authentic videos ($\partial P(V = 0 \mid B = 0)/\partial d_{T|T} > 0$) and the improved detection of inauthentic videos ($\partial P(V = 0 \mid B = 0)/\partial d_{F|F} > 0$). When detection is as good as random, $d_{F|F} = d_{T|T} = 1/2$, the probability that a video is inauthentic given one's belief that it is reduces to $\pi_F$. That is, the viewer must rely entirely on prior beliefs that the video is true. It follows that improvements in detection can lead to more accurate beliefs among viewers. In the first treatment, we estimate the detection rate of false videos, that is $d_{F|F} = P(B = 0 \mid V = 0)$.

Second, the model can be used to highlight both the potential benefits and pitfalls of content warnings. Consider a social media platform that decides to implement content warnings $C$ on uploaded videos that are deemed likely to be false. Suppose further that these warnings are imperfect (or perceived to be imperfect): while inauthentic videos are more likely to be flagged as deepfakes, authentic videos are now sometimes flagged as potential deepfakes. Content warnings therefore improve one margin of detection ($\partial d_{F|F}/\partial C > 0$) while degrading the other ($\partial d_{T|T}/\partial C < 0$). The net effect of content warnings on the probability of the viewer holding correct beliefs is therefore ambiguous, and depends on whether improvements from the former outweigh the costs of the latter.

In the second treatment, we test whether content warnings improve detection. We also provide more suggestive evidence that increased knowledge of deepfakes is correlated with more scepticism in online content, in the sense that more authentic videos are erroneously identified as inauthentic, i.e. $d_{T|T}(C = 1) < d_{T|T}(C = 0)$. Note that the presence of content warnings could in principle decrease detection abilities for authentic videos for two reasons: (i) the specific video in question erroneously has a content warning, or (ii) the presence of content warnings on the platform reduces detection abilities for authentic videos more generally, that is, even for videos without content warnings. The second effect highlights the possibility that content warnings could engender wider scepticism in all online content.

# 4. Experimental design

Our design focuses on two key questions surrounding the public's capacity to detect deepfakes with and without warnings. First, do people note anything out of the ordinary when they encounter a deepfake in a natural environment? Second, when directly warned they will see at least one deepfake in a set of videos, are people able to tell if a specific video is real or fake? We address these questions using a survey experiment with three experimental arms.

## 4.1. Detection with no content warnings

In the *No Deepfake, No Content Warning* (C1) arm, participants watch five unaltered videos before being asked whether they have noted anything out of the ordinary. In the *Presence of Deepfake, No Content Warning* (T1) arm, participants watch four of these same five unaltered videos plus a deepfake (which is fourth in the sequence), then answer the same series of questions. In both cases, if individuals answer 'yes' to noticing something out of the ordinary, they are further asked to indicate in which specific video(s) they thought something amiss, and to provide a short explanation of why. The comparison between C1 and T1 serves to answer the first question about deepfake detection without warnings: if people are alert to irregularities in deepfake videos in natural settings, we would expect to see a meaningful difference between these two groups in the proportion of participants reporting something out of the ordinary.

**Table 1.** Descriptive statistics.

|  | C1 mean | T1 versus C1 | T2 versus C1 |
|---|---|---|---|
| age | 45.82 (0.883) | −0.512 (1.251) | −1.108 (1.235) |
| female | 0.507 (0.026) | 0.069 (0.037) | 0.049 (0.037) |
| aware of deepfakes | 0.357 (0.025) | 0.004 (0.036) | 0.065 (0.036) |
| proficient with social media (1–10) | 6.025 (0.141) | −0.138 (0.195) | 0.148 (0.189) |
| internet use (1–10) | 6.416 (0.129) | −0.19 (0.186) | −0.109 (0.177) |
| familiar with actor (1–10) | 7.219 (0.135) | 0.021 (0.193) | −0.05 (0.185) |
| observations | 361 | 354 | 365 |

Notes: Difference in means in T1 and T2 relative to C1. Results are from an OLS regression of the row variable on an indicator for T1 and T2 for columns 2 and 3, respectively. Standard errors are in parentheses below the estimates.

## 4.2. Detection with content warnings

How do content warnings affect capacity for detection of false content? We address this question with a second treatment arm: *Presence of Deepfake, Content Warning* (T2). Participants in this arm watch the same set of videos as those in the *Presence of Deepfake, No Content Warning* (T1) treatment arm—one of which is a deepfake. However, before watching the videos, they are briefly informed what deepfakes are (manipulated videos that use deep learning artificial intelligence to make fake videos that appear real) and told that at least one of the five videos they will see is a deepfake. After watching all five videos in sequence, they are then asked to select which video(s) they believe are fake. Only those participants who select the deepfake and only the deepfake are counted as having correctly distinguished the fake video from the genuine content.

## 4.3. Videos

We make use of a deepfake video of the American actor Tom Cruise created and made public by the VFX artist Chris Ume. The clip is shown alongside a series of authentic video clips of Mr Cruise from publicly available YouTube channels. To control for past familiarity, all participants also watch a 1 min excerpt of an interview with Mr Cruise to provide a baseline acquaintance with the actor's appearance and speech patterns. All six videos can be viewed in our electronic supplementary material.

## 4.4. Procedure

We recruit a sample ($n = 1093$) of UK-based participants through Lucid Marketplace, which provides subject pools balanced on key demographics. Past research evaluating Lucid's data quality suggests the platform is in line with other online sample providers and is 'suitable for evaluating many social scientific theories' [18]. Participants complete a Qualtrics-based survey. We exclude any potential participants who are under 18, not residents of the UK, or are unwilling to provide informed consent to participate. Participants are randomized into one of the three experimental treatments. In line with our pre-registration plan, all participants must pass an attention check to be included in the study. We also remove subjects who have seen the deepfake video previously. This removes five respondents from T1 and 8 from T2. In line with our ethical procedure, all participants who view a deepfake are subsequently debriefed on which of the videos they viewed was inauthentic.

## 4.5. Descriptive statistics

Table 1 presents descriptive statistics for all three treatment groups. The first column reports the mean value of various characteristics in C1. Randomization implies that participants in all three groups should be similar in both observed (and unobserved) attributes. Columns 2 and 3 provide a check on

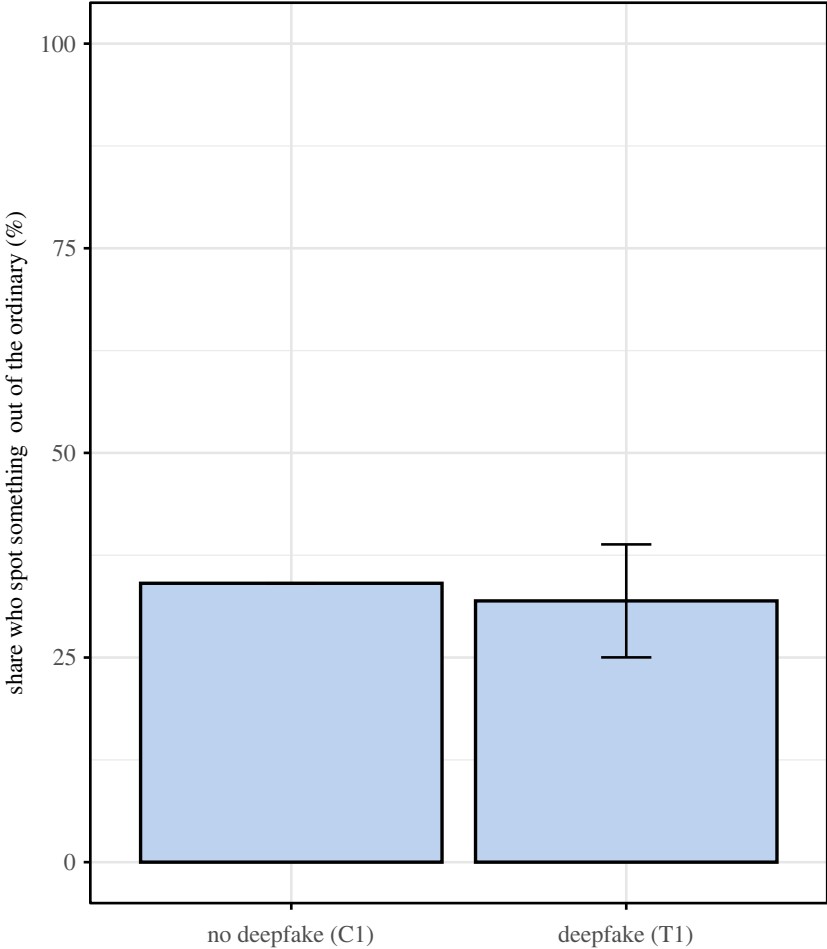

**Figure 1.** Manual detection of deepfakes with no content warnings. *Notes:* Whiskers show 95% confidence intervals calculated from a regression of the indicator outcome variable on an indicator for being in the treatment group using robust standard errors.

randomization balance by showing the difference in means for observed characteristics for T1 and T2 relative to C1. Differences in T1 and T2 relative to C1 are small and not statistically significant at the 5 per cent level, indicating balance across experimental conditions. The average age of participants is around 45, roughly half are female, and around 36% were previously aware of deepfakes.

# 5. Results

The topline results of our experiment are summarized in figures 1 and 2. Figure 1 shows that, without a content warning, individuals are no more likely to spot something out of the ordinary when exposed to one deepfake and four authentic videos, compared with a control group who saw only authentic videos. Figure 2 depicts the results of the second treatment, in which participants are warned at least one of the videos they will see is a deepfake. It shows the distribution of videos identified as a deepfake by participants, wherein the fourth video in the sequence (marked DF) is the deepfake. Only 21.6% of participants correctly identify the deepfake video as the only inauthentic video, while nearly half erroneously select more than one video as a deepfake, suggesting the warning may have increased scepticism in all videos, irrespective of their veracity. Thus, with or without a content warning, most individuals are unable to manually discern a deepfake from a genuine video.

## 5.1. Detection with no content warnings

The first treatment aims to answer the question: do people spot something amiss when they encounter a deepfake without a content warning? Participants are randomized into two conditions, *No Deepfake, No*

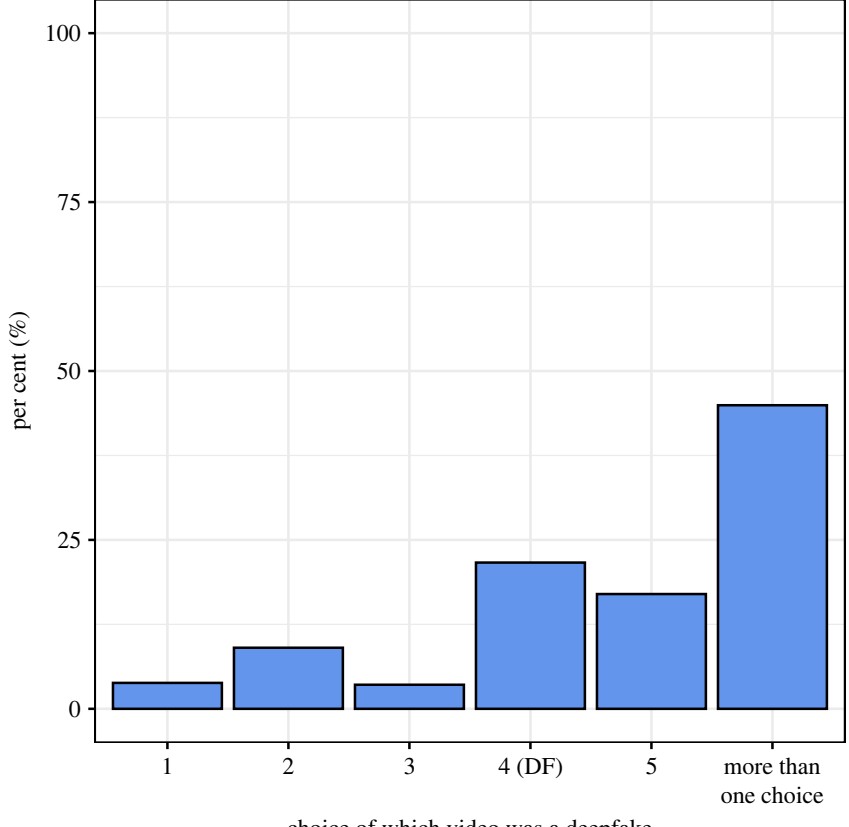

**Figure 2.** Manual detection of deepfakes with a content warning. *Notes:* Distribution of videos identified as deepfakes by participants who are warned they will see at least one deepfake in a set of five videos. Video four is the deepfake. The first five columns show the distribution of choices among those who chose only one video. The final group consists of those selecting more than one video.

*Content Warning* (C1) and *Presence of Deepfake, No Content Warning* (T1). Participants in C1 watch five authentic videos, while those in T1 watch four of the same authentic videos and a deepfake. All participants are then asked if they have found anything to be 'out of the ordinary' in the videos they watched. If participants respond *Yes* to this question, they are then asked to indicate the video(s) in which they found something out of the ordinary.

### 5.1.1. Spotting something out of the ordinary

Table 2 presents results from regressing an indicator for detecting something out of the ordinary from a set of videos on a treatment indicator for whether one of the five videos was a deepfake. Specifically, we regress

$$y_i = \alpha + \beta T_i + x_i' \delta + \varepsilon_i,$$

where $y_i$ is an indicator for reporting something is out of the ordinary in the set of videos viewed; $T_i$ is an indicator for viewing a set of videos that include the deepfake; $x_i$ is a vector of individual characteristics; and $\varepsilon_i$ is an unobserved error term. The coefficient of interest, $\beta$, measures the adjusted mean difference between C1 and T1 in detecting something out of the ordinary. Standard errors for all regressions are robust to heteroscedasticity.

Column 1 presents regression results with no covariates. In the *No Deepfake, No Content Warning* (C1) control arm, where all videos were authentic, only 34.1% of participants report having noted anything out of the ordinary. In the *Presence of Deepfake, No Content Warning* (T1) treatment arm, where one video was a deepfake, the fraction of participants reporting anything out of the ordinary was in fact slightly lower, at 32.9%. The difference in detection is small and not statistically significant at the 5 per cent level.

**Table 2.** OLS regressions: manual detection without content warnings.

| | dependent variable: | |
| | out of the ordinary | |
| | (1) | (2) |
|---|---|---|
| treatment (T1) | −0.022 (0.035) | −0.016 (0.035) |
| female | | −0.035 (0.036) |
| age × 1/10 | | −0.053*** (0.011) |
| aware of deepfakes | | 0.116*** (0.039) |
| proficient with social media | | 0.0005 (0.043) |
| high level of internet use | | 0.025 (0.040) |
| familiar with actor (0–10) | | 0.003 (0.007) |
| constant | 0.341*** (0.025) | 0.531*** (0.090) |
| observations | 715 | 694 |
| adjusted $R^2$ | −0.001 | 0.056 |

Notes: $*p < 0.1$; $**p < 0.05$; $***p < 0.01$. *Aware of deepfakes* is a binary variable equal to 1 if the participant was aware of deepfakes prior to participating in the experiment. *Proficiency with social media* is a binary variable equal to one if the participant's self-report is higher than the sample median for the question: 'On a scale of 0–10 where 10 is very proficient and 0 is not proficient at all, how proficient do you consider yourself in navigating social media platforms?' Similarly, *High level of internet use* is a binary variable equal to one if the participant's self-report is higher than the sample median for the question: 'On a scale of 0–10 where 10 is a great deal and 0 is none at all, how much time do you spend on the Internet outside of work-related commitments on an average day?' Standard errors are robust to heteroscedasticity.

Column 2 adds a set of covariates to the regression. As expected, the estimated treatment effect remains close to zero. However, the probability of reporting something out of the ordinary does vary with observable characteristics. Younger participants and those who were previously aware of deepfakes are more likely to report something is out of the ordinary. In particular, a 10-year reduction in age is associated with a 5.3 percentage point increase in reporting something is out of the ordinary. Those who were previously aware of deepfake technology were 11.6 percentage points more likely to report something out of the ordinary.

Around one-third of participants in both the control and the treatment groups identified something as out of the ordinary in the videos they viewed. These participants were asked further to identify which video (or videos) struck them as out of the ordinary. The most frequently chosen video in both treatment and control groups was the second video, which was in fact authentic. However, in the treatment group, the second most commonly chosen video was the deepfake. Together, this provides some suggestive evidence that detection may be better than random guessing but far from perfect, even among the selected sample of people who spot something out of the ordinary and encounter a deepfake. For the full distribution of videos identified as out of the original by this selected sample, see the appendix (figure 3).

### 5.1.2. Heterogeneous treatment effects

Younger subjects and those aware of deepfake technology are more likely to report that something is out of the ordinary. This raises a natural question: are these groups better able to detect the presence of an inauthentic video, or do they simply exhibit a higher base-line level of scepticism for online content? To answer this question, we regress

$$y_i = \alpha + \gamma H_i + \beta(T_i \times H_i) + \varepsilon_i \,,$$

where $H_i$ is an indicator for the dimension of heterogeneity (e.g. having prior awareness of deepfakes, or being younger than the median age). The coefficient $\beta$ measures the difference in reporting something out

**Table 3.** Heterogeneous treatment effect without content warnings.

| | dependent variable: | |
| --- | --- | --- |
| | out of the ordinary | |
| | (1) | (2) |
| aware of deepfakes | 0.144*** (0.048) | |
| aware of deepfakes × treatment (T1) | 0.066 (0.062) | |
| below median age | | 0.212*** (0.043) |
| below median age × treatment (T1) | | −0.020 (0.053) |
| constant | 0.266*** (0.021) | 0.229*** (0.022) |
| observations | 715 | 715 |
| adjusted $R^2$ | 0.032 | 0.044 |

Notes: *$p < 0.1$; **$p < 0.05$; ***$p < 0.01$. *Aware of deepfakes* is a binary variable equal to one if the participant was aware of deepfakes prior to participating in the experiment, and zero otherwise. *Below median age* is a binary variable equal to one if the participant is below the median age in the sample and zero otherwise. Standard errors are robust to heteroscedasticity.

of the ordinary between the subgroups in C1 and T1 for whom $H_i = 1$. If these subgroups are better able to detect something out of the ordinary, then $\beta > 0$. The coefficient $\gamma$ measures the difference in means in C1 between those with $H_i = 0$ and $H_i = 1$ of reporting something out of the ordinary; this measures baseline difference in scepticism towards online content.

Table 3 presents the results. In column 1, $H_i$ is a binary variable which equals 1 if $i$ reports having a prior awareness of deepfakes. In column 2, $H_i = 1$ if participant $i$ is below the median age in the sample, and 0 otherwise. The results do not show a statistically significant difference in detecting something out of the ordinary between those in C1 and T1 who were previously aware of deepfake technology and those who were below the median age. These subgroups, however, do have a higher baseline probability of reporting something out of the ordinary (which is consistent with results in table 2). Overall, these results show that certain subgroups are more likely to express scepticism over the authenticity of online content, but they are no more or less likely to report this when actually encountering a deepfake.

## 5.2. Detection with content warnings

Experimental participants are not more likely to spot something out of the ordinary when viewing a deepfake in a natural setting. Do content warnings enable detection? In the *Presence of Deepfake, Content Warning* (T2) treatment arm, participants received the following warning: 'On the following pages are a series of five additional videos of Mr Cruise, at least one of which is a deepfake video.' They were then asked to identify the video(s) they believed to be a deepfake; only one video was in fact a deepfake.

Figure 2 presents the results. The first five columns show the distribution of choices conditional on choosing only one video. The final group consists of those who selected more than one video. There are two main takeaways. First, the majority of participants (78.4%) were unable to correctly identify the deepfake as the only inauthentic video. Among those who selected only one video, participants were more likely to incorrectly identify one of the four genuine videos as a deepfake (60.7%) than to choose the correct video (39.3%). The distribution of choices, conditional on selecting a single video, shows that participants are somewhat more likely to correctly identify the deepfake than any other individual video. Overall, this is suggestive evidence of imperfect manual detection abilities. These results are broadly consistent with findings from Groh *et al*. [8], who find that human detection is better than random guessing and in line with leading detection algorithms with 65% accuracy. Note that with a 65% detection rate for both authentic and inauthentic videos, correctly identifying the authenticity of five consecutive videos, with no mistakes, has a probability of $0.65^5 = 0.116$ (assuming each trial is independent).

**Table 4.** Heterogeneity analysis with content warnings.

| | dependent variable: detection | | | | | | | | |
|---|---|---|---|---|---|---|---|---|---|
| | (1) | (2) | (3) | (4) | (5) | (6) | (7) | (8) | (9) |
| obvious: not | 0.061 (0.051) | | | | | | | | 0.068 (0.059) |
| obvious: don't know | 0.047 (0.054) | | | | | | | | 0.072 (0.060) |
| confidence: low | | −0.009 (0.069) | | | | | | | −0.082 (0.082) |
| confidence: medium | | −0.022 (0.052) | | | | | | | −0.087 (0.057) |
| female | | | −0.033 (0.044) | | | | | | −0.019 (0.046) |
| aware of deepfakes | | | | −0.015 (0.044) | | | | | 0.018 (0.050) |
| age × 1/10 | | | | | 0.035*** (0.013) | | | | 0.033*** (0.014) |
| proficient with social media | | | | | | −0.031 (0.046) | | | 0.020 (0.053) |
| high level of internet use | | | | | | | −0.099** (0.045) | | −0.074 (0.051) |
| familiar with actor | | | | | | | | 0.001 (0.044) | 0.004 (0.047) |
| constant | 0.179*** (0.036) | 0.231*** (0.044) | 0.235*** (0.033) | 0.223*** (0.029) | 0.062 (0.058) | 0.226*** (0.026) | 0.249*** (0.028) | 0.217*** (0.031) | 0.107 (0.094) |
| observations | 365 | 365 | 365 | 365 | 364 | 365 | 365 | 350 | 349 |
| adjusted $R^2$ | −0.002 | −0.005 | −0.001 | −0.002 | 0.017 | −0.002 | 0.010 | −0.003 | 0.007 |

Notes: $*p < 0.1$; $**p < 0.05$; $***p < 0.01$. For definitions of the variables, see notes in table 2. In columns 1–8, we report the coefficient of a regression of correct detection against a categorical variables. All coefficient are relative to an omitted category. Standard errors are robust to heteroscedasticity.

The second takeaway is that a large share of participants (45%) identified more than one video as a deepfake, despite the fact that there was only one deepfake. This implies that these participants incorrectly identified one or more authentic videos as a deepfake. The high share of participants selecting multiple videos provides suggestive evidence that content warnings may come at the cost of generating greater general scepticism, which spreads to authentic content. Policymakers should account for this potential cost when evaluating the efficacy of moderating online content.

### 5.2.1. Heterogeneity analysis

Are some groups of people better able to detect deepfakes than others? Table 4 shows results on how correct detection varies with observable individual characteristics. Each column examines a different dimension of heterogeneity, and the final column includes all of them in a single specification. Remarkably, the results show very few observable characteristics are correlated with correct detection. The only characteristic which is positively correlated with detection is age, where *older* participants are more likely to correctly identify the deepfake. In particular, an increase in 10 years of age is associated with an 3.3 percentage point increase in detection. By contrast, prior awareness of deepfakes, self-reported confidence in detection,[1] answering whether the deepfake was 'obvious', proficiency with social media, high internet use, and familiarity with the actor, were not associated with better or worse detection. Overall, the majority of participants, regardless of their observed characteristics, were not able to correctly identify the single deepfake.

## 6. Conclusion

Even as deepfakes have begun to reach a broad audience, the technology continues to evolve. These advances are only making it easier and less costly to produce higher-quality deepfakes—which can now be made to a convincing standard using publicly available programs [19]. As such, it seems likely that instances of such videos circulating on prominent social media platforms will increase in coming years. These realities necessitate the development of technological tools for detecting deepfakes, but also a better understanding of the technology's capacity to deceive.

To that end, our research makes two main contributions: first, we show that in natural browsing contexts, individuals are unlikely to note something unusual when they encounter a deepfake. This aligns with some previous findings indicating individuals struggle to detect high-quality deepfakes. Second, we present results on the effect of content warnings on detection, showing that the majority of individuals are still unable to spot a deepfake from a genuine video, even when they are told that at least one video in a series of videos they will view has been altered. Successful content moderation—for example, with specific videos flagged as fake by social media platforms—may therefore depend not on enhancing individuals' ability to detect irregularities in altered videos on their own, but instead on fostering trust in external sources of content authentication (particularly automated systems for deepfake detection).

As the public becomes more aware of the existence and deceptiveness of deepfakes, it is possible there will be growing scepticism towards online videos. Moreover, increasingly accurate and frequent detection of false videos by content moderators could have the unintended effect of undermining public trust in video media, irrespective of the veracity of an individual video. Future research might examine the importance of this potential challenge faced by content moderators and policymakers, and seek to advance our understanding of how individual characteristics such as ideological predisposition may affect trust in content moderation efforts.

**Data accessibility.** All supplementary materials, including data, code, and videos for replication, can be found at: https://github.com/patrick-vu/deepfakes.

**Declaration of AI use.** We have used AI-assisted technologies in creating this article.

**Authors' contributions.** All authors gave final approval for publication and agreed to be held accountable for the work performed therein.

**Conflict of interest declaration.** We declare we have no competing interests.

**Funding.** Our research was funded by The Royal Society.

---

[1]We create a categorical variable from a 0–10 self-reported scale of confidence in detection with three possibilities: low confidence (0–2); medium confidence (3–7); and high confidence (8–10).

# Appendix A

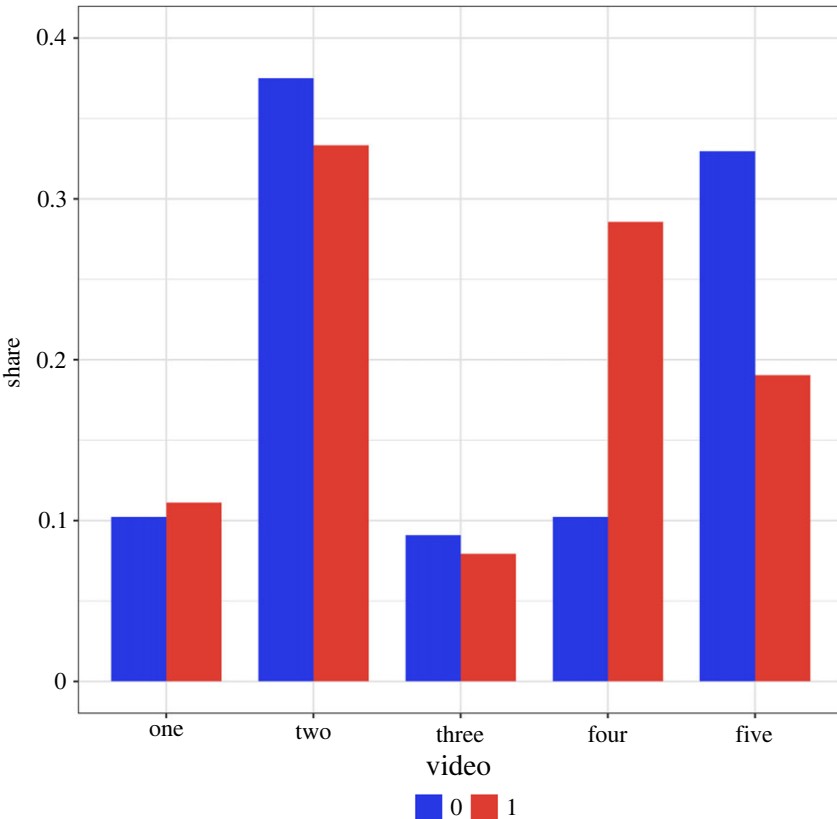

**Figure 3.** Distribution of videos selected as 'out of the ordinary'. *Notes:* The distribution of videos identified as out of the ordinary by participants in the control and first experimental treatment conditions. The sample is the subset of participants who indicated having observed something out of the ordinary. These participants were asked to identify which videos were out of the ordinary, with the option to select more than one video. The blue bars represent the control group, who viewed five authentic videos. The red bars represent the treatment group, who viewed four authentic videos and one deepfake (the fourth video).

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
