## [Peer Review File · Royal Society Open Science]

Review History

RSOS-220254.R0 (Original submission)

Review form: Reviewer 1

Is the manuscript scientifically sound in its present form?

Yes

Are the interpretations and conclusions justified by the results?

Yes

Is the language acceptable?

Yes

Do you have any ethical concerns with this paper?

No

Have you any concerns about statistical analyses in this paper?

Yes

Recommendation?

Accept with minor revision (please list in comments)

Comments to the Author(s)

The paper seems to be a first attempt at experimentally observing the impact of direct warnings on detecting deepfakes.

Good aspects:

- + basic structure of the paper.
- + experiments and statistical analysis

Minor issues:

* The authors should include some discussion about the need for a policy on content moderation. Besides policy, the authors should discuss other strategies to alert users to deepfake misinformation.

* I am curious that if the authors had created their own deepfake video instead of a publically available video, would it have impacted the results?

Typo: Page 6, Line 26: repetition of word by

Review form: Reviewer 2

Is the manuscript scientifically sound in its present form?

No

Are the interpretations and conclusions justified by the results?

No

Is the language acceptable?

No

Do you have any ethical concerns with this paper?

No

Have you any concerns about statistical analyses in this paper?

Yes

Recommendation?

Major revision is needed (please make suggestions in comments)

Comments to the Author(s)

Do Content Warnings Help People Spot a Deepfake? Evidence from Two Experiments

The current manuscript titled "Do Content Warnings Help People Spot a Deepfake? Evidence from Two Experiments" presents experimental findings on two main questions: First, how accurately can people detect a deepfake video when people are not suspecting that they are potentially being shown a deepfake? Second, how accurately can people detect a deepfake when prompted that one of the videos that they are shown is a deepfake? The authors collect data to answer these questions by recruiting 1,093 participants in the UK from Lucid, collecting data with Qualtrics, and using 5 real videos and one deepfake of Tom Cruise as experimental stimuli. Participants are randomly assigned to a control group, C, with 5 real videos, a treatment group, T1, with 4 real videos and one deepfake, and a second treatment group, T2, with 4 real videos and one deepfake where participants are told that one of the 5 videos is a deepfake before they see any of the videos. The authors present OLS regressions to show average treatment effects. They do not find statistically significant effects on "spot[ing] something out of the ordinary" for randomized assignment to T1 (4 real Tom Cruise videos and 1 deepfake) compared to

randomized assignment to C (5 real Tom Cruise videos). The authors claim “individuals who received no content warning correctly identify the deepfake in about 10% of cases, while those who are warned the content they see may be altered are twice as likely to detect it.” This claim about differences in T1 and T2 is not justified given how the researchers devised the experiment; T2 participants are forced to respond which video is a deepfake whereas T1 participants have a choice. Complete guessing in T2 would lead to the same result, which suggests these results are a product of the task design not the deception detection capability.

Overall, this paper addresses an extremely important topic, but there are multiple problems that need to be addressed before this paper should be published.

1. This paper claims “to present the first evaluation of how warning labels affect capacity for detection, by telling participants at least one of the videos they are to see is a deepfake and observing the proportion of respondents who correctly identify the altered content.” The experimental method for creating a reasonably ecologically valid setting by showing a number of videos and asking if any were out of the ordinary is a great contribution, but this is not the first paper to show how warning labels affect capacity for detection. For example, Pennycook et al 2020 and Epstein et al 2021 are papers that show effects of warning labels on fake news detection. Moreover, Groh et al 2020 in PNAS focuses on deepfake detection in a setting where participants are specifically tasked with distinguishing real from fake and are as such forewarned that videos may be deepfakes. Moreover, I suggest the authors to take a look at this editorial from Nature Human Behavior on self-claims of priority: <https://www.nature.com/articles/s41562-021-01068-x>
2. The language in this paper needs more precision. On line 24, 36, and 39 of page 1, there are three examples of using the word “will” where “may” is much more appropriate. This is not meant to be pedantic. Will indicates certainty whereas may allows for uncertainty, and each of these sentences presents consequences as more certain than they are. This issue continues throughout the article.
3. The authors cite technical tools as “proving effective” and cite one of the thousands of algorithmic deepfake detection papers with no clear justification. The 98.2% accuracy cited is deceptive because it refers to a model evaluated on a specific dataset and not videos writ large. Dolhasky et al presents the largest deepfake dataset to date and find that the best model in 2019 was 65% accurate on the DFDC dataset. Accuracies need to be contextualized because otherwise the readers might misunderstand how accurate algorithmic models can really be on real-world.
4. The authors claim “little is understood about how content warnings... will affect individuals capacity to detect deepfakes.” But the authors do not mention how Ternovski et al (or the other papers mentioned above) also examines warnings.
5. The authors claim “findings in Experiment 1 confirm past research showing people struggle to identify deepfakes from genuine videos without a warning.” But, past research shows this is not as clear as the authors make it appear. See Groh et al 2022 in PNAS and Groh et al 2022 on arXiv <https://arxiv.org/pdf/2202.12883.pdf> for nuances about how good people are at detecting deepfakes.
6. The description of the experimental design is awkward. The paper describes two experiments and claims the treatment in E1 is the control in E2. Why not describe this as one experiment with two treatment conditions?
7. In T1 and T2, can the authors add a description of the position of the deepfake in the 5 videos? Is it the fifth? Is it randomized?
8. In T2 on page 6, authors say participants are “told that at least one of the five videos they will see is a deepfake.” Why use “will” instead of “may”? With “may”, the authors could have also included a fourth arm with this warning and 5 real videos to make this a 2x2 factorial design experiment to handle the issue around T1 and T2 being incomparable due to the forced choice in T2.
9. Can the authors link to the pre-registration?
10. In table 1, the authors present t-values. I’m not sure why these aren’t standard errors or p-values.

11. On page 9, the authors write “the difference is statistically insignificant.” The authors conduct a test of statistical significance, and they do not find statistical significance. This is different than showing “statistical insignificance” (which is not really a thing) because absence of evidence is not evidence of absence.
12. Table 2, The Treatment (T1) could be re-labeled as Presence of Deepfake for clarity.
13. On page 11, the authors refer to Table 4 but I think they mean Table 3.
14. On page 13, the authors write “Experiment 1 shows that people are not likely to spot something out of the ordinary when viewing deepfakes” but the authors only show one deepfake.
15. In the conclusion, the authors write “This creates a tension” but it’s unclear what the tension is between. Is the tension between moderation, trust, and misinformation? This point could use more elaboration.

The methodology for evaluating deepfakes by asking about spotting anything unusual is a nice contribution to the deepfake detection literature, and this paper addresses an important question. However, there are many aspects this paper needs to address before it should be published.

This review is for Royal Society Open Science and is part of a “Peer Review Transparency Pilot” where I have been asked to share my name alongside my review, so I am also sharing it here for full transparency.

Decision letter (RSOS-220254.R0)

Dear Mr Lewis

The Editors assigned to your paper RSOS-220254 "Do Content Warnings Help People Spot a Deepfake? Evidence from Two Experiments" have now received comments from reviewers and would like you to revise the paper in accordance with the reviewer comments and any comments from the Editors. Please note this decision does not guarantee eventual acceptance.

Please submit your revised manuscript and required files (see below) no later than 21 days from today's (ie 10-May-2022) date. Note: the ScholarOne system will 'lock' if submission of the revision is attempted 21 or more days after the deadline. If you do not think you will be able to meet this deadline please contact the editorial office immediately.

Please note article processing charges apply to papers accepted for publication in Royal Society Open Science (<https://royalsocietypublishing.org/rsos/charges>). Charges will also apply to

papers transferred to the journal from other Royal Society Publishing journals, as well as papers submitted as part of our collaboration with the Royal Society of Chemistry (<https://royalsocietypublishing.org/rsos/chemistry>). Fee waivers are available but must be requested when you submit your revision (<https://royalsocietypublishing.org/rsos/waivers>).

on behalf of Dr Bruno Rossion (Associate Editor) and Nick Pearce (Subject Editor)
openscience@royalsociety.org

Associate Editor Comments to Author (Dr Bruno Rossion):

Associate Editor: 1

Comments to the Author:

Two reviewers find the paper interesting but have provided useful comments to revise the current version. The second reviewer in particular has provided extensive comments to take into account very seriously for revision of the paper. The revised paper will be sent to the same reviewers, possibly a third reviewer. Without adequate revisions, the paper will not be acceptable for publication in RSOS.

Reviewer comments to Author:

Reviewer: 1

Comments to the Author(s)

The paper seems to be a first attempt at experimentally observing the impact of direct warnings on detecting deepfakes.

Good aspects:

- + basic structure of the paper.
- + experiments and statistical analysis

Minor issues:

* The authors should include some discussion about the need for a policy on content moderation. Besides policy, the authors should discuss other strategies to alert users to deepfake misinformation.

* I am curious that if the authors had created their own deepfake video instead of a publically available video, would it have impacted the results?

Typo: Page 6, Line 26: repetition of word by

Reviewer: 2

Comments to the Author(s)

Do Content Warnings Help People Spot a Deepfake? Evidence from Two Experiments

The current manuscript titled "Do Content Warnings Help People Spot a Deepfake? Evidence from Two Experiments" presents experimental findings on two main questions: First, how accurately can people detect a deepfake video when people are not suspecting that they are potentially being shown a deepfake? Second, how accurately can people detect a deepfake when prompted that one of the videos that they are shown is a deepfake? The authors collect data to answer these questions by recruiting 1,093 participants in the UK from Lucid, collecting data with Qualtrics, and using 5 real videos and one deepfake of Tom Cruise as experimental stimuli.

Participants are randomly assigned to a control group, C, with 5 real videos, a treatment group, T1, with 4 real videos and one deepfake, and a second treatment group, T2, with 4 real videos and one deepfake where participants are told that one of the 5 videos is a deepfake before they see any of the videos. The authors present OLS regressions to show average treatment effects. They do not find statistically significant effects on “spot[ing] something out of the ordinary” for randomized assignment to T1 (4 real Tom Cruise videos and 1 deepfake) compared to randomized assignment to C (5 real Tom Cruise videos). The authors claim “individuals who received no content warning correctly identify the deepfake in about 10% of cases, while those who are warned the content they see may be altered are twice as likely to detect it.” This claim about differences in T1 and T2 is not justified given how the researchers devised the experiment; T2 participants are forced to respond which video is a deepfake whereas T1 participants have a choice. Complete guessing in T2 would lead to the same result, which suggests these results are a product of the task design not the deception detection capability.

Overall, this paper addresses an extremely important topic, but there are multiple problems that need to be addressed before this paper should be published.

1. This paper claims “to present the first evaluation of how warning labels affect capacity for detection, by telling participants at least one of the videos they are to see is a deepfake and observing the proportion of respondents who correctly identify the altered content.” The experimental method for creating a reasonably ecologically valid setting by showing a number of videos and asking if any were out of the ordinary is a great contribution, but this is not the first paper to show how warning labels affect capacity for detection. For example, Pennycook et al 2020 and Epstein et al 2021 are papers that show effects of warning labels on fake news detection. Moreover, Groh et al 2020 in PNAS focuses on deepfake detection in a setting where participants are specifically tasked with distinguishing real from fake and are as such forewarned that videos may be deepfakes. Moreover, I suggest the authors to take a look at this editorial from Nature Human Behavior on self-claims of priority: <https://www.nature.com/articles/s41562-021-01068-x>
2. The language in this paper needs more precision. On line 24, 36, and 39 of page 1, there are three examples of using the word “will” where “may” is much more appropriate. This is not meant to be pedantic. Will indicates certainty whereas may allows for uncertainty, and each of these sentences presents consequences as more certain than they are. This issue continues throughout the article.
3. The authors cite technical tools as “proving effective” and cite one of the thousands of algorithmic deepfake detection papers with no clear justification. The 98.2% accuracy cited is deceptive because it refers to a model evaluated on a specific dataset and not videos writ large. Dolhasky et al presents the largest deepfake dataset to date and find that the best model in 2019 was 65% accurate on the DFDC dataset. Accuracies need to be contextualized because otherwise the readers might misunderstand how accurate algorithmic models can really be on real-world.
4. The authors claim “little is understood about how content warnings... will affect individuals capacity to detect deepfakes.” But the authors do not mention how Ternovski et al (or the other papers mentioned above) also examines warnings.
5. The authors claim “findings in Experiment 1 confirm past research showing people struggle to identify deepfakes from genuine videos without a warning.” But, past research shows this is not as clear as the authors make it appear. See Groh et al 2022 in PNAS and Groh et al 2022 on arXiv <https://arxiv.org/pdf/2202.12883.pdf> for nuances about how good people are at detecting deepfakes.
6. The description of the experimental design is awkward. The paper describes two experiments and claims the treatment in E1 is the control in E2. Why not describe this as one experiment with two treatment conditions?
7. In T1 and T2, can the authors add a description of the position of the deepfake in the 5 videos? Is it the fifth? Is it randomized?
8. In T2 on page 6, authors say participants are “told that at least one of the five videos they will see is a deepfake.” Why use “will” instead of “may”? With “may”, the authors could have also included a fourth arm with this warning and 5 real videos to make this a 2x2 factorial design

experiment to handle the issue around T1 and T2 being incomparable due to the forced choice in T2.

9. Can the authors link to the pre-registration?

10. In table 1, the authors present t-values. I'm not sure why these aren't standard errors or p-values.

11. On page 9, the authors write "the difference is statistically insignificant." The authors conduct a test of statistical significance, and they do not find statistical significance. This is different than showing "statistical insignificance" (which is not really a thing) because absence of evidence is not evidence of absence.

12. Table 2, The Treatment (T1) could be re-labeled as Presence of Deepfake for clarity.

13. On page 11, the authors refer to Table 4 but I think they mean Table 3.

14. On page 13, the authors write "Experiment 1 shows that people are not likely to spot something out of the ordinary when viewing deepfakes" but the authors only show one deepfake.

15. In the conclusion, the authors write "This creates a tension" but it's unclear what the tension is between. Is the tension between moderation, trust, and misinformation? This point could use more elaboration.

The methodology for evaluating deepfakes by asking about spotting anything unusual is a nice contribution to the deepfake detection literature, and this paper addresses an important question. However, there are many aspects this paper needs to address before it should be published.

This review is for Royal Society Open Science and is part of a "Peer Review Transparency Pilot" where I have been asked to share my name alongside my review, so I am also sharing it here for full transparency.

===PREPARING YOUR MANUSCRIPT===

If you have been asked to revise the written English in your submission as a condition of publication, you must do so, and you are expected to provide evidence that you have received language editing support. The journal would prefer that you use a professional language editing service and provide a certificate of editing, but a signed letter from a colleague who is a fluent speaker of English is acceptable. Note the journal has arranged a number of discounts for authors

using professional language editing services
(<https://royalsociety.org/journals/authors/benefits/language-editing/>).

===PREPARING YOUR REVISION IN SCHOLARONE===

Author's Response to Decision Letter for (RSOS-220254.R0)

See Appendix A.

RSOS-220254.R1 (Revision)

Review form: Reviewer 2

Is the manuscript scientifically sound in its present form?

No

Are the interpretations and conclusions justified by the results?

No

Is the language acceptable?

No

Do you have any ethical concerns with this paper?

No

Have you any concerns about statistical analyses in this paper?

No

Recommendation?

Major revision is needed (please make suggestions in comments)

Comments to the Author(s)

The results reported in the re-written abstract need to be contextualized and clarified to avoid readers misinterpreting. In particular, (a) the deepfake video used as a stimulus is not only produced via "deep learning AI to create fake" but it is also the product of a highly skilled visual effects artist using standard visual effects tools (b) there is only one deepfake video in this experiment and the use of plural deepfakes overstates the generalization (c) the deepfake Tom Cruise video has been seen by tens of millions of people, so it's reasonable to refer to it directly as the Tom Cruise deepfake video (d) the authors claim to conduct an experiment "to detect deepfakes with the naked eye" but that is not a precise characterization of this experiment because this experiment directly asks people if anything is out of the ordinary (it's an experiment to reveal if people identify a deepfake as out of the ordinary) and it's important to clarify this point because the literature on when people are asked to detect deepfakes demonstrates people are much better than random at detecting deepfakes (e.g. see Groh et al in PNAS and Barari et al preprint, which the authors cite in the lit review) (e) Figure 2 and the 21.6% claim in the abstract seem misleading because it seems (though I can't tell because of how this is worded) that a higher percentage of participants identified video 4 as deepfake but these people aren't included in the 21.6 number because they also thought another video was also a deepfake. Please add clarity about the accuracy rate and false positive rate, and (f) it is unclear how the possible implication in the last sentence is connected with the data.

In line 17 of page 10, you may consider including human accuracy rates from the Groh et al PNAS paper given that this submission is also focused on human detection abilities and Groh et al PNAS evaluated on a large sample from the DFDC competition mentioned.

Lines 40-45 of page 10 are fantastic and characterize the research nicely!

Line 50 of page 10 about rendering faith in judgments of content moderators is unclear. Who are the content moderators? Why does it make them essential? If you want to make this point, you'll need to expand and provide evidence and an explanation.

The literature review does not fairly characterize Groh et al PNAS contribution to the human detection of deepfakes space. In particular, Groh et al PNAS demonstrates that in two separate settings: (a) two alternative forced choice between a real and fake video (b) a single video (sometimes real and sometimes fake), humans are significantly better than random guessing and depending on how one measures it also better or comparable to the leading algorithms. Likewise, the literature review should include Groh et al preprint <https://arxiv.org/abs/2202.12883> which contributes to this literature and further demonstrates that people are significantly better than random chance at detecting deepfakes when asked directly.

Line 31 on page 12, what is elusive about manual detection? Do the authors mean imperfect instead of elusive?

The authors should identify and link to the 5 videos being used because this paper is based on only 5 real stimuli and one deepfake and it's possible that something out of the ordinary appeared in the video in the control that is not a deepfake but led people to make such an assessment. I'd like to see a robustness analysis to demonstrate that no single non-deepfake video is being identified as out of the ordinary at high rates.

In the media summary, I'm not sure how the authors can justify they have a representative sample of the British public – in the body of the paper, they say they use Lucid on key demographic characteristics. Also, this research does not show people are unable to discern deepfakes from real videos. Instead, this shows a more nuanced result that people do not generally identify a single Tom Cruise deepfake produced by VFX artist Chris Ume as out of the ordinary. In fact, past research on hundreds of deepfakes shows that people are significantly better than random guessing. Finally, the claim that content warnings increase distrust is also not supported by the data. Distrust is not simply false positive identification; distrust could be measured by subjective reporting but it is unclear how the data and experimental design supports claims on distrust

Decision letter (RSOS-220254.R1)

Dear Mr Lewis

The Editors assigned to your paper RSOS-220254.R1 "Deepfake Detection With and Without Content Warnings" have now received comments from reviewers and would like you to revise the paper in accordance with the reviewer comments and any comments from the Editors. Please note this decision does not guarantee eventual acceptance.

Please submit your revised manuscript and required files (see below) no later than 21 days from today's (ie 20-Oct-2022) date. Note: the ScholarOne system will 'lock' if submission of the revision is attempted 21 or more days after the deadline. If you do not think you will be able to meet this deadline please contact the editorial office immediately.

on behalf of Dr Bruno Rossion (Associate Editor) and Nick Pearce (Subject Editor)
openscience@royalsociety.org

Associate Editor Comments to Author (Dr Bruno Rossion):

Comments to the Author:

The paper should be revised (major revision). The reviewer has provided an extensive series of comments to revise the paper. Please also include the 6 videos in the data

Reviewer comments to Author:

Reviewer: 2

Comments to the Author(s)

The results reported in the re-written abstract need to be contextualized and clarified to avoid readers misinterpreting. In particular, (a) the deepfake video used as a stimulus is not only produced via "deep learning AI to create fake" but it is also the product of a highly skilled visual effects artist using standard visual effects tools (b) there is only one deepfake video in this experiment and the use of plural deepfakes overstates the generalization (c) the deepfake Tom Cruise video has been seen by tens of millions of people, so it's reasonable to refer to it directly as the Tom Cruise deepfake video (d) the authors claim to conduct an experiment "to detect deepfakes with the naked eye" but that is not a precise characterization of this experiment because this experiment directly asks people if anything is out of the ordinary (it's an experiment to reveal if people identify a deepfake as out of the ordinary) and it's important to clarify this point because the literature on when people are asked to detect deepfakes demonstrates people are much better than random at detecting deepfakes (e.g. see Groh et al in PNAS and Barari et al preprint, which the authors cite in the lit review) (e) Figure 2 and the 21.6% claim in the abstract seem misleading because it seems (though I can't tell because of how this is worded) that a higher percentage of participants identified video 4 as deepfake but these people aren't included in the 21.6 number because they also thought another video was also a deepfake. Please add clarity about the accuracy rate and false positive rate, and (f) it is unclear how the possible implication in the last sentence is connected with the data.

In line 17 of page 10, you may consider including human accuracy rates from the Groh et al PNAS paper given that this submission is also focused on human detection abilities and Groh et al PNAS evaluated on a large sample from the DFDC competition mentioned.

Lines 40-45 of page 10 are fantastic and characterize the research nicely!

Line 50 of page 10 about rendering faith in judgments of content moderators is unclear. Who are the content moderators? Why does it make them essential? If you want to make this point, you'll need to expand and provide evidence and an explanation.

The literature review does not fairly characterize Groh et al PNAS contribution to the human detection of deepfakes space. In particular, Groh et al PNAS demonstrates that in two separate settings: (a) two alternative forced choice between a real and fake video (b) a single video (sometimes real and sometimes fake), humans are significantly better than random guessing and depending on how one measures it also better or comparable to the leading algorithms. Likewise, the literature review should include Groh et al preprint <https://arxiv.org/abs/2202.12883> which contributes to this literature and further demonstrates that people are significantly better than random chance at detecting deepfakes when asked directly.

Line 31 on page 12, what is elusive about manual detection? Do the authors mean imperfect instead of elusive?

The authors should identify and link to the 5 videos being used because this paper is based on only 5 real stimuli and one deepfake and it's possible that something out of the ordinary appeared in the video in the control that is not a deepfake but led people to make such an assessment. I'd like to see a robustness analysis to demonstrate that no single non-deepfake video is being identified as out of the ordinary at high rates.

In the media summary, I'm not sure how the authors can justify they have a representative sample of the British public - in the body of the paper, they say they use Lucid on key demographic characteristics. Also, this research does not show people are unable to discern deepfakes from real videos. Instead, this shows a more nuanced result that people do not generally identify a single Tom Cruise deepfake produced by VFX artist Chris Ume as out of the ordinary. In fact, past research on hundreds of deepfakes shows that people are significantly better than random guessing. Finally, the claim that content warnings increase distrust is also not supported by the data. Distrust is not simply false positive identification; distrust could be measured by subjective reporting but it is unclear how the data and experimental design supports claims on distrust

===PREPARING YOUR MANUSCRIPT===

If you have been asked to revise the written English in your submission as a condition of publication, you must do so, and you are expected to provide evidence that you have received language editing support. The journal would prefer that you use a professional language editing service and provide a certificate of editing, but a signed letter from a colleague who is a fluent speaker of English is acceptable. Note the journal has arranged a number of discounts for authors using professional language editing services (<https://royalsociety.org/journals/authors/benefits/language-editing/>).

===PREPARING YOUR REVISION IN SCHOLARONE===

RSOS-231214.R0 (Original submission)

Review form: Reviewer 2

Is the manuscript scientifically sound in its present form?

Yes

Are the interpretations and conclusions justified by the results?

Yes

Is the language acceptable?

Yes

Do you have any ethical concerns with this paper?

No

Have you any concerns about statistical analyses in this paper?

No

Recommendation?

Accept as is

Comments to the Author(s)

The authors have nicely addressed the previous comments and this manuscript looks good.

Review form: Reviewer 3

Is the manuscript scientifically sound in its present form?

Yes

Are the interpretations and conclusions justified by the results?

Yes

Is the language acceptable?

Yes

Do you have any ethical concerns with this paper?

Yes

Have you any concerns about statistical analyses in this paper?

Yes

Recommendation?

Accept with minor revision (please list in comments)

Comments to the Author(s)

Thank you for the opportunity to review "Deepfake Detection With and Without Content Warnings." This paper uses an online survey experiment, conducted in the UK, to evaluate (i) individuals' ability to detect deepfake videos, and (ii) the efficacy of content warnings in improving discernment of deepfake versus authentic videos. Overall, the authors find that, absent content warnings, members of the public exposed to a deepfake are no more likely to

discern anything out of the ordinary than members of the public exposed only to authentic videos. In addition, they find that generalized content warnings do little to improve people's ability to distinguish between deepfakes and authentic videos.

This type of experimental work is both timely and important, and it has important implications for the design of warning labels on social media platforms, as well as digital literacy and other educative interventions aimed at improving people's knowledge and awareness of deepfakes and other artificial intelligence tools. Despite these valuable contributions, however, I have several questions about the framing and research design (described in further detail below). If these issues can be addressed, I would be more than happy to re-evaluate the paper's contribution.

Overall Comments

- My main concern is about the **research design** — specifically, the structure of the content warning. If I understood the design correctly, the content warning informed respondents that at least one of the five videos they would be shown was a deepfake.
 - I question, however, to what extent this content warning mimics strategies that platforms or practitioners would actually use. Namely, my sense is that there are two broad categories of warning labels: (i) specific labels applied to individual pieces of content, and (ii) general warnings that highlight the existence of deepfakes, without providing any information about whether individual pieces of content are or are not deepfakes.
 - I struggle to grasp what the real-world analog of the treatment would be, since it seems unlikely that people would ever encounter a warning that some sample of content would contain a deepfake.
 - As such, I would appreciate more discussion from the authors about why they selected this particular approach to labeling, as well as what they think can be learned from using this type of label that cannot be learned when studying more naturalistic forms of warning labels.
- Along these lines, I was curious about the decision to have the warning communicate that there was "at least one" deepfake in the stimulus set.
 - This language does seem to imply the presence of multiple deepfakes, which could account for the patterns observed in the study.
- In addition, I was a bit confused by the "out of the ordinary" dependent variable used for the C1 and T1 treatments, as it seems to be theoretically disconnected from the broader research question of deepfake detection.
 - Based on the open-ended probe, what kinds of explanations did respondents provide for why certain videos seemed amiss? Looking at Figure A1, what was going on with the (authentic) video 2 that made so many respondents flag this video as unusual?
 - It feels notable that approximately one-third of respondents in both the control group and the "no warning" group indicated seeing something out of the ordinary — a figure that, at least in my estimation, feels somewhat high! This suggests to me that the dependent variable isn't necessarily calibrated to the theoretical quantities of interest.
- Related to this former point, my interpretation of the research design is that respondents in the "warning" condition were asked about a different dependent variable than respondents in the other two conditions.
 - This strikes me as an unfortunate missed opportunity, insofar as it impedes the ability for the authors to make any comparisons across experimental conditions. For instance, are people more likely to say they detect something out of the ordinary when shown a deepfake content warning? (Presumably yes)
 - If you had measured both sets of dependent variables for all respondents (or elicited some consistent measure of accuracy/authenticity across conditions), this might have enabled estimation of the efficacy of the warning label. Without this comparison, though, it's hard to know how the warning label compared to the case where no warnings were provided.

- Given that this experimental and empirical approach varies sharply from how other scholars have sought to estimate the effects of labels, it feels important for the authors to provide some rationale behind their decision – and engage clearly with the question of how the T2 results can be interpreted in the absence of a baseline.

- I had some questions about the decision to make the stimulus set entirely about Tom Cruise.
 - First is the question of why the authors chose the Tom Cruise deepfake for the study – a question I think the authors should engage with at least briefly. Although I understand the logic of including a relatively innocuous deepfake that was unlikely to cause harm or affect political/personal outcomes, it does seem like the kind of content people might simply not care about (meaning that they might not detect it as a deepfake, but even if it went undetected, it might have minimal harms for society at large).
 - Second is the question of why the authors chose to show **only** Tom Cruise videos. If the goal was to simulate a naturalistic video feed (e.g., on Facebook or TikTok), it seems implausible that people would only encounter information about the exact same person or issue. How does asking people to evaluate a bunch of videos about the same person affect the kinds of inferences you're able to draw?

Minor Comments

- In general, I do not think novelty should be the primary criterion for determining a paper's contribution, but it does seem that this paper is walking in the footsteps of a wide body of existing research (much uncited) looking at deepfake detection.
 - I would appreciate the authors spending a bit of time indicating what unique insights their approach provides that other papers have not.
 - If this is largely meant as a replication or extension of previous work, that's not a problem – but as of now, it's unclear to me how the research was intended to build upon existing knowledge.
- One of the pre-treatment covariates in the survey was prior awareness of deepfakes. Where was this item positioned relative to the other items, and how concerned should we be that asking about deepfakes primed respondents to be attentive to false or inauthentic content?
- Table 3: I found the tables (or, perhaps, the model specifications) somewhat confusing, in that they seem to exclude the base term on the treatment indicator.
 - Namely, they include a base term on the moderator variable (e.g., age, prior awareness of deepfakes) but just include the interaction between the moderator and the treatment, without also including the base term on the treatment.
 - Perhaps I'm misunderstanding the model, but I was a bit confused about the rationale of excluding the main effect!
- In the conclusion, the authors suggest that people's inability to detect deepfakes (especially when not primed to think about this content) requires building trust in content moderators.
 - However, building on the comments in the reviewer memo, these results have very little to say about this question of trust. In fact, the argument seems to be that many respondents took the warning label at face value, but the warning label was too general to be useful.
 - So the takeaway for me isn't simply "build trust in content moderators" but rather to "design warning labels that provide targeted feedback on individual pieces of content."
- Page 4: "Our findings in the first experiment show..." and then "In the second treatment" – would clarify the first language, since there's only one study, just with multiple treatment arms.
 - And, arguably, there is one experiment and then one observational study, since the dependent variables are entirely disjoint across C1/T1 and T2.
- Page 5: "In all of the studies discussed, participants were issued a specific warning that they would view a deepfake"

- I don't believe this is true - in the Ternovski et al. (2022) study, they randomly assigned some people to receive a deepfake warning and others to receive no information about veracity.
- Similarly, Vaccari and Chadwick (2020) varied how much information they provided about the veracity of the Obama deepfake based on how the video was cut.
- It seems more accurate to say that the studies randomly varied the presence of warning labels – an approach the present study departs from in a notable and impactful way.

- Page 7: In the theoretic model, there is some discussion of what happens when warning labels are imperfect, such that authentic videos are sometimes flagged as potential deepfakes.

- Another scenario is that warning labels may imperfectly cover inauthentic videos, such that some deepfakes may evade detection (a not-unlikely scenario, given how the technology is continually evolving and learning from past attempts at detection)

- Should the effects of this type of imperfect warning system just be the inverse of the one laid out on page 7, or are there different considerations at play (given the structure of people's prior beliefs/likelihood of assuming information is true ex ante)?

- I would appreciate the authors providing some discussion (perhaps in an appendix) about the ethics of the work and any steps they took in the experiment to address the potential harms of showing people highly realistic misinformation.

Decision letter (RSOS-231214.R0)

Dear Dr Lewis

On behalf of the Editors, we are pleased to inform you that your Manuscript RSOS-231214 "Deepfake Detection With and Without Content Warnings" has been accepted for publication in Royal Society Open Science subject to minor revision in accordance with the referees' reports. Where applicable, please find the referees' comments along with any feedback from the Editors below my signature.

Please submit your revised manuscript and required files (see below) no later than 7 days from today's (ie 07-Nov-2023) date. Note: the ScholarOne system will 'lock' if submission of the revision is attempted 7 or more days after the deadline. If you do not think you will be able to meet this deadline please contact the editorial office immediately.

Kind regards,
Royal Society Open Science Editorial Office
Royal Society Open Science

on behalf of Dr Bruno Rossion (Associate Editor) and Nick Pearce (Subject Editor)
openscience@royalsociety.org

Associate Editor Comments to Author:

Thank you for the continued patience with the review of this paper: one of the reviewers of the original version of the paper has recommended acceptance, while a new reviewer for this iteration has some comments that we'd like you to address in a final version for submission. If you can supply a marked-up version of the revision and a clear point-by-point response, there should not be any need to return the paper to reviewers.

Reviewer comments to Author:

Reviewer: 1

Comments to the Author(s)

The authors have nicely addressed the previous comments and this manuscript looks good.

Reviewer: 2

Comments to the Author(s)

Thank you for the opportunity to review "Deepfake Detection With and Without Content Warnings." This paper uses an online survey experiment, conducted in the UK, to evaluate (i) individuals' ability to detect deepfake videos, and (ii) the efficacy of content warnings in improving discernment of deepfake versus authentic videos. Overall, the authors find that, absent content warnings, members of the public exposed to a deepfake are no more likely to discern anything out of the ordinary than members of the public exposed only to authentic videos. In addition, they find that generalized content warnings do little to improve people's ability to distinguish between deepfakes and authentic videos.

This type of experimental work is both timely and important, and it has important implications for the design of warning labels on social media platforms, as well as digital literacy and other educative interventions aimed at improving people's knowledge and awareness of deepfakes and other artificial intelligence tools. Despite these valuable contributions, however, I have several questions about the framing and research design (described in further detail below). If these issues can be addressed, I would be more than happy to re-evaluate the paper's contribution.

Overall Comments

- My main concern is about the **research design** – specifically, the structure of the content warning. If I understood the design correctly, the content warning informed respondents that at least one of the five videos they would be shown was a deepfake.
 - I question, however, to what extent this content warning mimics strategies that platforms or practitioners would actually use. Namely, my sense is that there are two broad categories of warning labels: (i) specific labels applied to individual pieces of content, and (ii) general warnings that highlight the existence of deepfakes, without providing any information about whether individual pieces of content are or are not deepfakes.
 - I struggle to grasp what the real-world analog of the treatment would be, since it seems unlikely that people would ever encounter a warning that some sample of content would contain a deepfake.
 - As such, I would appreciate more discussion from the authors about why they selected this particular approach to labeling, as well as what they think can be learned from using this type of label that cannot be learned when studying more naturalistic forms of warning labels.
- Along these lines, I was curious about the decision to have the warning communicate that there was "at least one" deepfake in the stimulus set.
 - This language does seem to imply the presence of multiple deepfakes, which could account for the patterns observed in the study.

- In addition, I was a bit confused by the “out of the ordinary” dependent variable used for the C1 and T1 treatments, as it seems to be theoretically disconnected from the broader research question of deepfake detection.
 - Based on the open-ended probe, what kinds of explanations did respondents provide for why certain videos seemed amiss? Looking at Figure A1, what was going on with the (authentic) video 2 that made so many respondents flag this video as unusual?
 - It feels notable that approximately one-third of respondents in both the control group and the “no warning” group indicated seeing something out of the ordinary – a figure that, at least in my estimation, feels somewhat high! This suggests to me that the dependent variable isn’t necessarily calibrated to the theoretical quantities of interest.

- Related to this former point, my interpretation of the research design is that respondents in the “warning” condition were asked about a different dependent variable than respondents in the other two conditions.
 - This strikes me as an unfortunate missed opportunity, insofar as it impedes the ability for the authors to make any comparisons across experimental conditions. For instance, are people more likely to say they detect something out of the ordinary when shown a deepfake content warning? (Presumably yes)
 - If you had measured both sets of dependent variables for all respondents (or elicited some consistent measure of accuracy/authenticity across conditions), this might have enabled estimation of the efficacy of the warning label. Without this comparison, though, it’s hard to know how the warning label compared to the case where no warnings were provided.
 - Given that this experimental and empirical approach varies sharply from how other scholars have sought to estimate the effects of labels, it feels important for the authors to provide some rationale behind their decision – and engage clearly with the question of how the T2 results can be interpreted in the absence of a baseline.

- I had some questions about the decision to make the stimulus set entirely about Tom Cruise.
 - First is the question of why the authors chose the Tom Cruise deepfake for the study – a question I think the authors should engage with at least briefly. Although I understand the logic of including a relatively innocuous deepfake that was unlikely to cause harm or affect political/personal outcomes, it does seem like the kind of content people might simply not care about (meaning that they might not detect it as a deepfake, but even if it went undetected, it might have minimal harms for society at large).
 - Second is the question of why the authors chose to show *only* Tom Cruise videos. If the goal was to simulate a naturalistic video feed (e.g., on Facebook or TikTok), it seems implausible that people would only encounter information about the exact same person or issue. How does asking people to evaluate a bunch of videos about the same person affect the kinds of inferences you’re able to draw?

Minor Comments

- In general, I do not think novelty should be the primary criterion for determining a paper’s contribution, but it does seem that this paper is walking in the footsteps of a wide body of existing research (much uncited) looking at deepfake detection.
 - I would appreciate the authors spending a bit of time indicating what unique insights their approach provides that other papers have not.
 - If this is largely meant as a replication or extension of previous work, that’s not a problem – but as of now, it’s unclear to me how the research was intended to build upon existing knowledge.

- One of the pre-treatment covariates in the survey was prior awareness of deepfakes. Where was this item positioned relative to the other items, and how concerned should we be that asking about deepfakes primed respondents to be attentive to false or inauthentic content?

- Table 3: I found the tables (or, perhaps, the model specifications) somewhat confusing, in that they seem to exclude the base term on the treatment indicator.
 - Namely, they include a base term on the moderator variable (e.g., age, prior awareness of deepfakes) but just include the interaction between the moderator and the treatment, without also including the base term on the treatment.
 - Perhaps I'm misunderstanding the model, but I was a bit confused about the rationale of excluding the main effect!

- In the conclusion, the authors suggest that people's inability to detect deepfakes (especially when not primed to think about this content) requires building trust in content moderators.
 - However, building on the comments in the reviewer memo, these results have very little to say about this question of trust. In fact, the argument seems to be that many respondents took the warning label at face value, but the warning label was too general to be useful.
 - So the takeaway for me isn't simply "build trust in content moderators" but rather to "design warning labels that provide targeted feedback on individual pieces of content."

- Page 4: "Our findings in the first experiment show..." and then "In the second treatment" - would clarify the first language, since there's only one study, just with multiple treatment arms.
 - And, arguably, there is one experiment and then one observational study, since the dependent variables are entirely disjoint across C1/T1 and T2.

- Page 5: "In all of the studies discussed, participants were issued a specific warning that they would view a deepfake"
 - I don't believe this is true - in the Ternovski et al. (2022) study, they randomly assigned some people to receive a deepfake warning and others to receive no information about veracity.
 - Similarly, Vaccari and Chadwick (2020) varied how much information they provided about the veracity of the Obama deepfake based on how the video was cut.
 - It seems more accurate to say that the studies randomly varied the presence of warning labels - an approach the present study departs from in a notable and impactful way.

- Page 7: In the theoretic model, there is some discussion of what happens when warning labels are imperfect, such that authentic videos are sometimes flagged as potential deepfakes.
 - Another scenario is that warning labels may imperfectly cover inauthentic videos, such that some deepfakes may evade detection (a not-unlikely scenario, given how the technology is continually evolving and learning from past attempts at detection)
 - Should the effects of this type of imperfect warning system just be the inverse of the one laid out on page 7, or are there different considerations at play (given the structure of people's prior beliefs/likelihood of assuming information is true ex ante)?

- I would appreciate the authors providing some discussion (perhaps in an appendix) about the ethics of the work and any steps they took in the experiment to address the potential harms of showing people highly realistic misinformation.

===PREPARING YOUR MANUSCRIPT===

one version should clearly identify all the changes that have been made (for instance, in coloured highlight, in bold text, or tracked changes);

===PREPARING YOUR REVISION IN SCHOLARONE===

- Ensure that your data access statement meets the requirements at <https://royalsociety.org/journals/authors/author-guidelines/#data>. You should ensure that you cite the dataset in your reference list. If you have deposited data etc in the Dryad repository, please only include the 'For publication' link at this stage. You should remove the 'For review' link.
- If you are requesting an article processing charge waiver, you must select the relevant waiver option (if requesting a discretionary waiver, the form should have been uploaded, see 'File upload' above).
- If you have uploaded any electronic supplementary (ESM) files, please ensure you follow the guidance at <https://royalsociety.org/journals/authors/author-guidelines/#supplementary-material> to include a suitable title and informative caption. An example of appropriate titling and captioning may be found at https://figshare.com/articles/Table_S2_from_Is_there_a_trade-off_between_peak_performance_and_performance_breadth_across_temperatures_for_aerobic_scope_in_teleost_fishes_/3843624.

Author's Response to Decision Letter for (RSOS-231214.R0)

See Appendix B.

Decision letter (RSOS-231214.R1)

Dear Mr Lewis:

I am pleased to inform you that your manuscript entitled "Deepfake Detection With and Without Content Warnings" is now accepted for publication in Royal Society Open Science.

Please remember to make any data sets or code libraries 'live' prior to publication, and update any links as needed when you receive a proof to check - for instance, from a private 'for review' URL to a publicly accessible 'for publication' URL. It is also good practice to add data sets, code and other digital materials to your reference list. At this stage, we ask that you please archive your GitHub code within the Zenodo repository: <https://guides.github.com/activities/citable-code/>. By doing this, a formal, citable DOI will be associated with your data record, and an open license (CC-BY preferred) can be applied to your data. Please send us an amended data access statement in the format below, and we can amend this for you.

"Data and relevant code for this research work are stored in GitHub: [GitHub URL here] and have been archived within the Zenodo repository: <https://doi.org/zenodo.....> [ref number].

Royal Society Open Science is a fully open access journal. A payment may be due before your article is published. Please note that, if the corresponding author of your paper is based at an institution covered by one of our Transformative Agreement deals, your fees may be covered by the deal – please check the list of eligible institutions

at <https://royalsociety.org/journals/authors/read-and-publish/read-publish-agreements/>. The Royal Society has partnered with Copyright Clearance Center's (CCC's) RightsLink service to allow authors to pay article processing charges or page charges. After your manuscript has been accepted, the corresponding author will receive an email from CCC with the subject "Please submit your article processing/open access charge(s)/page charges" inviting you to pay your charges or request an invoice. The email from CCC will come from the email domain @copyright.com (if you have any queries regarding fees, please see <https://royalsocietypublishing.org/rsos/charges> or contact authorfees@royalsociety.org). If you request an invoice, it will be sent to you from CCC.

It is important to be cautious about payment scams. If you receive an email or text message requesting payment and have any concerns, we recommend contacting us through our website, rather than clicking on any links. The Royal Society will never ask you to make a direct payment.

on behalf of Dr Bruno Rossion (Associate Editor) and Professor Nick Pearce (Subject Editor).

Associate Editor Dr Bruno Rossion Comments to Author:

<https://www.facebook.com/RoyalSocietyPublishing/>

Andrew Lewis
University College, Oxford
High Street, Oxford, OX1 4BH
andrew.lewis@univ.ox.ac.uk

15 July, 2022

Dear Dr Rossion,

Thank you for giving us the opportunity to revise and resubmit our paper. We are grateful to both reviewers for taking the time to assess our manuscript and provide valuable feedback. We have carefully considered the reviewers' comments, and incorporated their suggestions into our updated manuscript. Below is a point-by-point response to each comment.

Reviewer 1 Comments

Comment 1: *The authors should include some discussion about the need for a policy on content moderation. Besides policy, the authors should discuss other strategies to alert users to deepfake misinformation.*

Authors' response: Thanks for this suggestion. In the Introduction, we have included a discussion about content provenance, which is one of the main potential strategies (along with algorithmic detection) for combating deepfakes. This addition can be found in the third paragraph of the Introduction. In the Conclusion, we also discuss some of the potential challenges that policymakers may face in developing content moderation policies, and the need for future research on how to best mitigate these.

Comment 2: *I am curious that if the authors had created their own deepfake video instead of a publically available video, would it have impacted the results?*

Authors' response: It's an interesting question. The somewhat mixed results around deepfake detection (highlighted by Reviewer 2) seem to suggest that detection depends on a number of video characteristics, for example, the video's quality, the identity of the speaker(s), and the content of what is being said. Our article is interested in high-quality deepfakes with a well-known figure and neutral content. We expect that our results would be similar with different deepfake videos — including ones we might produce — that share these characteristics. Of course, using a publicly available deepfake allows for the potential that participants will have seen the video before, which we attempted to control for by excluding participants who self-report having seen the deepfake previously.

Comment 3: *Typo: Page 6, Line 26: repetition of word by.*

Authors' response: Thanks for pointing this out. We have corrected this.

Reviewer 2 Comments

Comment in opening paragraph: *The authors claim “individuals who received no content warning correctly identify the deepfake in about 10% of cases, while those who are warned the content they see may be altered are twice as likely to detect it.” This claim about differences in T1 and T2 is not justified given how the researchers devised the experiment; T2 participants are forced to respond which video is a deepfake whereas T1 participants have a choice. Complete guessing in T2 would lead to the same result, which suggests these results are a product of the task design not the deception detection capability.*

Authors' response: We agree with the reviewer here that these two groups are not directly comparable, and as such we have decided to drop the direct comparison from the paper (changes pertaining to this point have been made in the abstract, introduction, results, and concluding sections). As the reviewer notes, participants in T1 and T2 face different choice environments, with those in T2 forced to select at least one video as a deepfake, while those in T1 only do so if they first report noticing something out of the ordinary. Thus, the group in T1 is a selected sample of more alert people. Ultimately this is a product of our design, which favoured a focus on the “natural context” component of T1 over direct comparability between treatments.

Rather than making this head-to-head comparison, we now simply present the proportion of participants correctly identifying the deepfake in each condition. For T2, we now present results with the full distribution of video choices, as well as an overall summary of correct versus incorrect choice. We note that because participants could select more than 1 video — which roughly 45% did — the distribution of complete guessing would not be 20% per video (as it would be if participants could select only one video). The histogram summarising choices in T2 should make clearer that video choice is not random (i.e., equally distributed across videos).

Comment 1: *This paper claims “to present the first evaluation of how warning labels affect capacity for detection, by telling participants at least one of the videos they are to see is a deepfake and observing the proportion of respondents who correctly identify the altered content.” The experimental method for creating a reasonably ecologically valid setting by showing a number of videos and asking if any were out of the ordinary is a great contribution, but this is not the first paper to show how warning labels affect capacity for detection. For example, Pennycook et al 2020 and Epstein et al 2021 are papers that show effects of warning labels on fake news detection. Moreover, Groh et al 2020 in PNAS focuses on deepfake detection in a setting where participants are specifically tasked with distinguishing real from fake and are as such forewarned that videos may be deepfakes. Moreover, I suggest the authors to take a look at this editorial from Nature Human Behavior on self-claims of priority:*

<https://www.nature.com/articles/s41562-021-01068-x>

Authors' response: We thank the reviewer for this comment and for pointing us to the editorial from Nature Human Behavior. We have removed claims of priority and instead have tried simply to identify the contribution of our design. We have also added additional references, including

those suggested, to our paper, adding in a review of the broader literature on content warnings pertaining to and beyond deepfakes. These changes can be found in the third paragraph of the revised literature review. Where claims of priority had been made, changes rendered can be seen in the marked-up version of our revised manuscript.

Comment 2: *The language in this paper needs more precision. On line 24, 36, and 39 of page 1, there are three examples of using the word “will” where “may” is much more appropriate. This is not meant to be pedantic. Will indicates certainty whereas may allows for uncertainty, and each of these sentences presents consequences as more certain than they are. This issue continues throughout the article.*

Authors’ response: We agree with the reviewer’s emphasis on using precise language for claims. We have made changes to this effect throughout the paper, including in the specific instances mentioned in this comment. Each individual change can be seen in the marked-up version of our manuscript in the tracked changes.

Comment 3: *The authors cite technical tools as “proving effective” and cite one of the thousands of algorithmic deepfake detection papers with no clear justification. The 98.2% accuracy cited is deceptive because it refers to a model evaluated on a specific dataset and not videos writ large. Dolhasky et al presents the largest deepfake dataset to date and find that the best model in 2019 was 65% accurate on the DFDC dataset. Accuracies need to be contextualized because otherwise the readers might misunderstand how accurate algorithmic models can really be on real-world.*

Authors’ response: We agree with the reviewer’s assessment. We have changed the sentence to better reflect the range of algorithmic accuracy, particularly the roughly 65% efficacy figure for detection *in the wild* using the DFDC dataset, which we now include with the suggested Dolhansky et al citation. See the third paragraph of the Introduction for the relevant change.

Comment 4: *The authors claim “little is understood about how content warnings... will affect individuals capacity to detect deepfakes.” But the authors do not mention how Ternovski et al (or the other papers mentioned above) also examines warnings.*

Authors’ response: We have included a more complete review of previous research on content warnings — for both deepfake and non-deepfake misinformation. These changes have been reflected in the revised literature review. As highlighted in the paper, we also discuss some of the differences between direct content warnings (e.g., *this video is a deepfake*) and generalised warnings, such as that provided by Ternovski and co-authors, which they describe as an “information warning about the existence of deepfakes” broadly, with no direct indication a specific video is fake. This enables us to better situate our research within the context of what has already been done.

Comment 5: *The authors claim “findings in Experiment 1 confirm past research showing people struggle to identify deepfakes from genuine videos without a warning.” But, past research shows this is not as clear as the authors make it appear. See Groh et al 2022 in PNAS and Groh*

et al 2022 on arXiv <https://arxiv.org/pdf/2202.12883.pdf> for nuances about how good people are at detecting deepfakes.

Authors' response: We appreciate the reviewer's assertion that past evidence on detection ability is more mixed than we had cast it. We've included the suggested citation and revised the literature review to reflect this wider range of findings on detection ability, situating our findings therein. See the first paragraph of the literature review for the specific change.

Comment 6: *The description of the experimental design is awkward. The paper describes two experiments and claims the treatment in E1 is the control in E2. Why not describe this as one experiment with two treatment conditions?*

Authors' response: We agree and have recast the description of the experimental design as a single experiment with one control arm (*No Deepfake, No Content Warning*) and two active treatment arms (*Presence of Deepfake, No Content Warning* and *Presence of Deepfake, Content Warning*). See sections 4.1 and 4.2 of the revised Experimental Design (Section 4) for the first instance of these changes, which are subsequently effected throughout the document.

Comment 7: *In T1 and T2, can the authors add a description of the position of the deepfake in the 5 videos? Is it the fifth? Is it randomized?*

Authors' response: Thanks for this suggestion. We have included a description of the (non-randomised) position of the deepfake in the second sentence of Section 4.1, which is always fourth among the five videos. Participants view all videos from start to finish before being asked any questions about their veracity, which we believe minimises the likelihood of order effects from non-randomisation. The below histogram shows distribution of choice amongst those spotting something out of the ordinary in C1. As can be seen, the 4th video (which in T1 and T2 is the deepfake) is not among the most selected in C1. We believe this provides some evidence that selection of the deepfake in T1 and T2 is not undermined by non-randomised order effects.

Comment 8: *In T2 on page 6, authors say participants are “told that at least one of the five videos they will see is a deepfake.” Why use “will” instead of “may”? With “may”, the authors could have also included a fourth arm with this warning and 5 real videos to make this a 2x2 factorial design experiment to handle the issue around T1 and T2 being incomparable due to the forced choice in T2.*

Authors’ response: We agree that using ‘may’ could have led to a more flexible design. Ultimately we were most interested in answering the question: *knowing* there is a fake video among a set of videos, can people detect which video is the deepfake? The use of ‘may’, while enabling direct comparison between T1 and T2, would leave this question unresolved.

Our concern with using ‘may’ would be that participants might estimate the probability of there being a deepfake as anywhere between zero and one. We would then have no way of discerning to what extent selection would be based on individuals’ priors. We do think providing participants with specific probabilities (e.g., there is a 25 / 50 / 75% chance one of the videos is a deepfake) might make for an interesting future design.

Comment 9: *Can the authors link to the pre-registration?*

Authors’ response: Yes, apologies for not having done so before. Our pre-registration can be accessed via AsPredicted at <https://aspredicted.org/sz8g6.pdf>.

Comment 10: *In table 1, the authors present t-values. I’m not sure why these aren’t standard errors or p-values.*

Authors' response: Following the reviewer's suggestion, the t-values in Table 1 have been changed to standard errors for consistency with other tables.

Comment 11: *On page 9, the authors write "the difference is statistically insignificant." The authors conduct a test of statistical significance, and they do not find statistical significance. This is different than showing "statistical insignificance" (which is not really a thing) because absence of evidence is not evidence of absence.*

Authors' response: Agree very much with this comment. The sentence now reads: "The difference in detection is small and not statistically significant at the 5 percent level." We have also changed descriptions of the results elsewhere in the article to better reflect both the size of differences (practical significance) and statistical significance (rather than 'insignificance'). For example, Section 4.5 Descriptive Statistics.

Comment 12: *Table 2, The Treatment (T1) could be re-labeled as Presence of Deepfake for clarity.*

Authors' response: We have followed this suggestion and given all experimental conditions more intuitive names. Further details are in our response to Comment 6 about revising the description of the experimental design.

Comment 13: *On page 11, the authors refer to Table 4 but I think they mean Table 3.*

Authors' response: Thanks for pointing that out. We've changed this.

Comment 14: *On page 13, the authors write "Experiment 1 shows that people are not likely to spot something out of the ordinary when viewing deepfakes" but the authors only show one deepfake.*

Authors' response: We have changed this to read '... when viewing a deepfake'.

Comment 15: *In the conclusion, the authors write "This creates a tension" but it's unclear what the tension is between. Is the tension between moderation, trust, and misinformation? This point could use more elaboration.*

Authors' response: We agree that this passage could do with greater clarity. We have reworded these sentences and replaced the word 'tension' with 'challenge'. The challenge we have in mind is that increasingly accurate (and frequent) detection of false content by moderators could inadvertently lead the public to have greater distrust of all videos, including authentic content.

Having thought carefully about and acted upon each of these comments, we believe our revised manuscript presents a much-improved version of our research. We again thank the editors and reviewers for their time and look forward to receiving a decision in due course.

Sincerely,

Andrew Lewis

On behalf of all authors.

Dr Andrew Lewis
University College, Oxford
High Street, Oxford, OX1 4BH
andrew.lewis@univ.ox.ac.uk

15 November, 2023

Dear Dr Rossion,

Thank you for your note in conjunction with this third round of reviewer comments. We are grateful to both reviewers for taking the time to assess our manuscript and provide valuable feedback. We have carefully considered these additional comments, and incorporated suggestions into our updated manuscript where appropriate. Below is a point-by-point response to each comment.

Editor Comments

Comment 1: Thank you for the continued patience with the review of this paper: one of the reviewers of the original version of the paper has recommended acceptance, while a new reviewer for this iteration has some comments that we'd like you to address in a final version for submission. If you can supply a marked-up version of the revision and a clear point-by-point response, there should not be any need to return the paper to reviewers.

Authors' response: Thank you for your note. We have replied in detail to each of the new reviewer's thoughtful comments below, with notes for corresponding changes made in-text. We appreciate your continued stewardship of this process, and hope to receive another decision in the near future.

Reviewer 1 Comments

The authors have nicely addressed the previous comments and this manuscript looks good.

Authors' response: Many thanks for this, we are grateful for the past and present rounds of comments which have brought our manuscript to the present form, which we agree is of a higher quality than the original.

Reviewer 2 Comments

Comment 1: My main concern is about the research design—specifically, the structure of the content warning. If I understood the design correctly, the content warning informed respondents that at least one of the five videos they would be shown was a deepfake. I question, however, to what extent this content warning mimics strategies that platforms or practitioners would actually use. Namely, my sense is that there are two broad categories of warning labels: (i) specific labels applied to individual pieces of content, and (ii) general warnings that highlight the existence of deepfakes, without providing any information about whether individual pieces of content are or are not

deepfakes. I struggle to grasp what the real-world analog of the treatment would be, since it seems unlikely that people would ever encounter a warning that some sample of content would contain a deepfake. As such, I would appreciate more discussion from the authors about why they selected this particular approach to labeling, as well as what they think can be learned from using this type of label that cannot be learned when studying more naturalistic forms of warning labels.

Authors' response: Thank you for these comments. Our design choice is connected to our primary purpose, which is to experimentally test whether people can reliably detect deepfakes. We agree that the content warning is not a direct analogue for the type one would expect to receive on social media. Whereas warnings on social media platforms have been (and will likely be in the future) attached to specific pieces of suspicious content (e.g., *Our moderators have determined that part or whole of this video have been digitally altered*), we do not apply this framework to the experiment for two primary reasons. First, this would entail attaching a warning to a specific video, either genuine or fake, and then asking people if they believed the video was a deepfake. While this would provide interesting insights into whether, for example, people believe content warnings, it would not answer our main question of interest, namely whether people can detect deepfakes from ordinary videos. For example, if after receiving a warning that a video was a deepfake a participant responded that they believed the video was a deepfake, would that tell us that participant had strong deepfake detection abilities, or that they had simply believed the warning? Second, if we did want to parse detection versus belief in warnings, such a design would require us to place warning labels on both real and fake videos in order to compare ultimate belief that a video was a fake after receiving a warning, which would violate the ethics policy against deception at our lab at Oxford. The purpose of our experiment is not so much to proxy a genuine social media experience as it is to answer the first necessary question in understanding the potential effects of deepfakes: can people tell they are fake?

Comment 2: Along these lines, I was curious about the decision to have the warning communicate that there was “at least one” deepfake in the stimulus set. This language does seem to imply the presence of multiple deepfakes, which could account for the patterns observed in the study.

Authors' response: Our reason for using “at least one” was to further isolate detection abilities. Past research has shown a great deal of false positive detection, which we are interested in. We believe that a respondent reporting a genuine video as a deepfake, regardless of the warning, unambiguously reflects poorly on their ability to tell genuine from fake videos apart, which is the primary purpose of our study.

Comment 3: In addition, I was a bit confused by the “out of the ordinary” dependent variable used for the C1 and T1 treatments, as it seems to be theoretically disconnected from the broader research question of deepfake detection. Based on the open-ended probe, what kinds of explanations did respondents provide for why certain videos seemed amiss? Looking at Figure A1, what was going on with the (authentic) video 2 that made so many respondents flag this video as unusual? It feels notable that approximately one-third of respondents in both the control group and the “no warning” group indicated seeing something out of the ordinary—a figure that, at least in my estimation, feels somewhat high! This suggests to me that the dependent variable isn't necessarily calibrated to the theoretical quantities of interest.

Authors' response: We appreciate you drawing attention to this distinction. This is, ultimately, a choice we made in the design to prioritise the *lack-of-warning* in the first two conditions, which does come at the expense of direct comparability. The purpose of using the “out of the ordinary” variable relates to the points made in Reply #1 above: for the control and Treatment 1, the *out of the ordinary* variable allows us to compare how alert people are to synthetic content in a more naturalistic environment — that is, without any warning that it may be fake. If we had said “one of these videos may be a deepfake,” we believe that would constitute something like a content warning, engendering alertness that may have otherwise been absent. This comparison answers the question: *without a content warning, does a deepfake video raise more suspicion of abnormality than a genuine video?* Thus there are two primary insights from these first conditions: first, when exposed to a deepfake, are people *alert* to synthetic content, and second, if it does catch their attention do they identify it correctly? That we observe no difference between the control (all genuine videos) and the treatment (containing a deepfake), with 1/3rd of participants in both conditions saying they thought something was out of the ordinary, indicates to us that the answer to this question is *no*.

While 1/3rd does seem a bit high, there is no external benchmark to which we can compare the 1/3rd number. The theoretical quantity of interest here is precisely whether the deepfake raises more suspicion than a genuine video, and thus strikes people as *out of the ordinary* — so we believe that it is well calibrated to answering that. Most of the responses as for why the genuine videos are perceived as out of the ordinary (sadly) pertain to the content of the video rather than the authenticity, despite the wording of the question which stated “thinking of the quality, rather than the content.” Given these videos were chosen in similar proportions between conditions, we nonetheless believe this operationalisation clearly answers *no* to the question of whether the deepfake itself, as a result of its inauthenticity, raises inordinate suspicion or stands out as obviously faked.

Comment 4: Related to this former point, my interpretation of the research design is that respondents in the “warning” condition were asked about a different dependent variable than respondents in the other two conditions. This strikes me as an unfortunate missed opportunity, insofar as it impedes the ability for the authors to make any comparisons across experimental conditions. For instance, are people more likely to say they detect something out of the ordinary when shown a deepfake content warning? (Presumably yes). If you had measured both sets of dependent variables for all respondents (or elicited some consistent measure of accuracy/authenticity across conditions), this might have enabled estimation of the efficacy of the warning label. Without this comparison, though, it’s hard to know how the warning label compared to the case where no warnings were provided. Given that this experimental and empirical approach varies sharply from how other scholars have sought to estimate the effects of labels, it feels important for the authors to provide some rationale behind their decision—and engage clearly with the question of how the T2 results can be interpreted in the absence of a baseline

Authors' response: As discussed in the reply above, we agree the two treatments are conceptually distinct and do not allow for direct comparison. We are open to the idea that a better operationalisation is possible. However, we do not agree with the premise that asking if something was perceived as out of the ordinary concurrently *with* a content warning would provide that better comparison — it is our view that, if you tell people something *is* out of the ordinary in one of the videos, subsequently asking them whether they believe something is out of the ordinary in one of those videos may not provide much practical insight into their independent alertness or detection abilities. The reviewer notes that the T2 results do not have a baseline. We agree and therefore

present this as suggestive empirical evidence. It is ultimately for the reader to decide whether they believe a 1/5 accuracy rate for detecting deepfakes constitutes *high* or *low* or *medium* detection ability.

Comment 5: I had some questions about the decision to make the stimulus set entirely about Tom Cruise. First is the question of why the authors chose the Tom Cruise deepfake for the study—a question I think the authors should engage with at least briefly. Although I understand the logic of including a relatively innocuous deepfake that was unlikely to cause harm or affect political/personal outcomes, it does seem like the kind of content people might simply not care about (meaning that they might not detect it as a deepfake, but even if it went undetected, it might have minimal harms for society at large).

Second is the question of why the authors chose to show *only* Tom Cruise videos. If the goal was to simulate a naturalistic video feed (e.g., on Facebook or TikTok), it seems implausible that people would only encounter information about the exact same person or issue. How does asking people to evaluate a bunch of videos about the same person affect the kinds of inferences you're able to draw?

Authors' response: Thanks for these considerations. The idea of showing the same individual across all videos was to try to control for as many factors as possible that could affect detection — allowing us to get more precise estimates. We are mainly interested in making inferences about detection. We wanted to be sure that any differences in detection would be related to the authenticity (or inauthenticity) of the videos, rather than other characteristics of the videos. We wanted to minimise variability between controls and treatments outside of the variation of the main independent variable(s). The trade-off is, as noted, that it mimics less well a naturalistic video feed.

The primary rationale for using the Tom Cruise deepfake was that, at the time we prepared the experiment, it was the highest quality deepfake in the public domain that included a well-known public figure (rather than an unknown individual). The second rationale is precisely because it is relatively innocuous. This is again to focus on detection. The interaction between perceptions of authenticity and more political content is a question we are very interested in, but outside the scope of this particular paper.

Comment 6: In general, I do not think novelty should be the primary criterion for determining a paper's contribution, but it does seem that this paper is walking in the footsteps of a wide body of existing research (much uncited) looking at deepfake detection. I would appreciate the authors spending a bit of time indicating what unique insights their approach provides that other papers have not. If this is largely meant as a replication or extension of previous work, that's not a problem—but as of now, it's unclear to me how the research was intended to build upon existing knowledge.

Authors' response: Thanks for this. We agree that novelty is not the main criterion for evaluating an academic contribution. We did previously explicitly call attention to what we see as our novel contribution in the paper itself, but were asked to remove that in a previous round of reviews.

Our paper addresses simply two questions: first, does inauthenticity *per se* elicit people's attention? Second, does a content warning enable an individual to judge a video's authenticity for themselves, to the extent that human capacity for detection will be sufficient? We appreciate the field is rapidly advancing. We do believe, however, that the most well-known and highly cited papers in this space remain the ones we discuss and build on.

Comment 7: One of the pre-treatment covariates in the survey was prior awareness of deepfakes. Where was this item positioned relative to the other items, and how concerned should we be that asking about deepfakes primed respondents to be attentive to false or inauthentic content?

Authors' response: Thanks for this. This was a post-treatment covariate — as were all covariates — so no reason to be concerned about priming in this instance.

This does however underscore our Reply #3 above, where we explain why we did not use a deepfake warning in C1 and T1, for fear of priming respondents to think about deepfakes where they otherwise would not have done — so we are indeed sensitive to the potential for priming and sought to minimise it (though there are always drawbacks to any design decision).

Comment 8: Table 3: I found the tables (or, perhaps, the model specifications) somewhat confusing, in that they seem to exclude the base term on the treatment indicator. Namely, they include a base term on the moderator variable (e.g., age, prior awareness of deepfakes) but just include the interaction between the moderator and the treatment, without also including the base term on the treatment. Perhaps I'm misunderstanding the model, but I was a bit confused about the rationale of excluding the main effect!

Authors' response: Thanks for this comment. The main effect could be included and the results would be unchanged. Our choice to specify the model in this way was simply to make the interpretation of the coefficients easier. For further explanation, note that a regression of a variable on a set of dummy variables will simply yield differences in means across particular groups. This holds for both our specification and the suggested specification. The mean comparison we are most interested in is the following: conditioning on being aware of deepfakes (or below median age in the second column), are those who view a deepfake more likely to detect something out of the ordinary? The answer to that question is given by the coefficient on the interaction in Table 3. If the model were, alternatively, to include a baseline for the treatment, then we would get the exact same number by summing the coefficient on the treatment indicator and the interaction. The downside of this approach (aside from having to do the arithmetic yourself) is that you don't get a standard error for the main mean difference you're interested in and thus cannot do standard hypothesis testing.

Comment 9: In the conclusion, the authors suggest that people's inability to detect deepfakes (especially when not primed to think about this content) requires building trust in content moderators. However, building on the comments in the reviewer memo, these results have very little to say about this question of trust. In fact, the argument seems to be that many respondents took the warning label at face value, but the warning label was too general to be useful.
- So the takeaway for me isn't simply "build trust in content moderators" but rather to "design warning labels that provide targeted feedback on individual pieces of content."

Authors' response: Thanks for this, we have sought to clarify. We agree that our results have little to say about *whether or not people trust content warnings*. What they do speak to is how important automatic detection systems (and related content warnings) will be in people's future determinations of content authenticity. The logic is as follows: if we wish to know whether a specific video is authentic or inauthentic with close to 100% certainty, and our ability to manually make this distinction is imperfect (i.e., <100%), we will necessarily need an external source of authentication to make this determination. Thus our question is not whether people trust content warnings — but how important people's trust sources of content authentication, will be in the future for separating deepfakes from genuine videos.

This understanding builds off both the model we present in the paper as well as those found in some past research, such as Fallis's (2021) epistemic threat of deepfakes model. We had a more robust discussion of this point in previous versions of the paper, but were asked to remove in previous reviews.

We have changed the summary of this in-text to read “external sources of authentication” rather than “content moderators,” which we hope clarifies our point. (Changes on p.3 and p.18).

Comment 10: Page 4: “Our findings in the first experiment show...” and then “In the second treatment” – would clarify the first language, since there's only one study, just with multiple treatment arms. And, arguably, there is one experiment and then one observational study, since the dependent variables are entirely disjoint across C1/T1 and T2.

Authors' response: Many thanks for catching this. This was an error on our part and we apologise for having missed it. The language on page 3 has been updated in-text to reflect this change, now reading “in the first treatment” rather than “in the first experiment.”

Comment 11: Page 5: “In all of the studies discussed, participants were issued a specific warning that they would view a deepfake”. I don't believe this is true – in the Ternovski et al. (2022) study, they randomly assigned some people to receive a deepfake warning and others to receive no information about veracity. Similarly, Vaccari and Chadwick (2020) varied how much information they provided about the veracity of the Obama deepfake based on how the video was cut. It seems more accurate to say that the studies randomly varied the presence of warning labels—an approach the present study departs from in a notable and impactful way.

Authors' response: Many thanks for the distinction. We have changed the wording in-text to read “some participants” rather than participants which may have implied all participants (p.4).

Comment 12: Page 7: In the theoretic model, there is some discussion of what happens when warning labels are imperfect, such that authentic videos are sometimes flagged as potential deepfakes.

- Another scenario is that warning labels may imperfectly cover inauthentic videos, such that some deepfakes may evade detection (a not-unlikely scenario, given how the technology is continually evolving and learning from past attempts at detection)
- Should the effects of this type of imperfect warning system just be the inverse of the one laid

out on page 7, or are there different considerations at play (given the structure of people's prior beliefs/likelihood of assuming information is true ex ante)?

Authors' response: We agree that automatic detection systems are and are likely to remain imperfect. While we are unsure about what different considerations may be at play with respect to beliefs and priors, we think that **all sources** of imperfect detection will impede perfect detection in the same direction.

Comment 13: I would appreciate the authors providing some discussion (perhaps in an appendix) about the ethics of the work and any steps they took in the experiment to address the potential harms of showing people highly realistic misinformation.

Authors' response: We agree that this line of research can present a number of potential ethical pitfalls. The research design was approved by the ethical review board at Oxford. Overall, we believe the potential harms in our work are rather limited given 1) the innocuous nature of the deepfake used, and 2) the lack of deception (no fake videos are presented as real, and all participants are debriefed after the treatment about which video was in fact fake). We have added a note in section 4.4 (p.8) to clarify that all participants who viewed a deepfake received this debrief.